# Spatial correlations constrain cellular lifespan and pattern formation in corneal epithelium homeostasis

Lior Strinkovsky[1], Evgeny Havkin[1], Ruby Shalom-Feuerstein[2,3], Yonatan Savir[1,3]*

[1]Department of Physiology, Biophysics and System Biology, Faculty of Medicine, Technion, Haifa, Israel; [2]Department of Genetics & Developmental Biology, Faculty of Medicine, Technion, Haifa, Israel; [3]The Rappaport Family Institute for Research in the Medical Sciences, Technion, Haifa, Israel

**Abstract** Homeostasis in adult tissues relies on the replication dynamics of stem cells, their progenitors and the spatial balance between them. This spatial and kinetic coordination is crucial to the successful maintenance of tissue size and its replenishment with new cells. However, our understanding of the role of cellular replicative lifespan and spatial correlation between cells in shaping tissue integrity is still lacking. We developed a mathematical model for the stochastic spatial dynamics that underlie the rejuvenation of corneal epithelium. Our model takes into account different spatial correlations between cell replication and cell removal. We derive the tradeoffs between replicative lifespan, spatial correlation length, and tissue rejuvenation dynamics. We determine the conditions that allow homeostasis and are consistent with biological timescales, pattern formation, and mutants phenotypes. Our results can be extended to any cellular system in which spatial homeostasis is maintained through cell replication.

*For correspondence:
yoni.savir@technion.ac.il

## Introduction

In adult tissues, stem cells and their progeny play a crucial role in maintaining homeostasis. Renewal of the tissue is due to progenitor cells that have limited replication capacity (*Watt and Hogan, 2000*). The interplay between stem cells and their progenitors with respect to replication, differentiation, and cellular hierarchy is not fully understood. For example, two opposing limiting models of stem cell replication have been proposed: A 'Hierarchical' model where stem cells are rare slow-dividing cells with longevity similar to the hematopoietic stem cell paradigm (*Orkin and Zon, 2008*) and a 'Equipotent' model where stem-cells are abundant equipotent cells that divide frequently and their loss is dictated by neutral drift (*Klein and Simons, 2011*; *Klein et al., 2010*; *Losick and Desplan, 2008*; *Lopez-Garcia et al., 2010*). Another lingering question is the role of spatial correlation between replication and removal in homeostasis. Some studies assume a long-range correlation between replication and cell removal, that is, as a cell replicates, the removed cell can be tens and even hundreds of cells away (*Lobo et al., 2016*; *Richardson et al., 2018*; *Park et al., 2019*; *Richardson et al., 2017*) and other studies, in different experimental systems, suggest short-range correlations between replication and removal (*Mesa et al., 2018*; *Miroshnikova et al., 2018*).

The cornea acts as a lens that focuses light into the eye, and serves as a barrier that protects the eye against external hazards or injury. Thus, maintaining its integrity and its continuous regeneration is crucial for proper vision in vertebrates (*Yazdanpanah et al., 2017*). It is now predominantly accepted that the regeneration of the corneal epithelium, in homeostasis, is due to limbal epithelial stem cells (LESCs) residing at the circumference of the cornea, the limbus, which separates the cornea from the conjunctiva (*Figure 1A*; *O'Callaghan and Daniels, 2011*; *Cotsarelis et al., 1989*; *Pellegrini et al., 1997*; *Lavker et al., 2004*; *Davanger and Evensen, 1971*; *Dziasko and Daniels,*

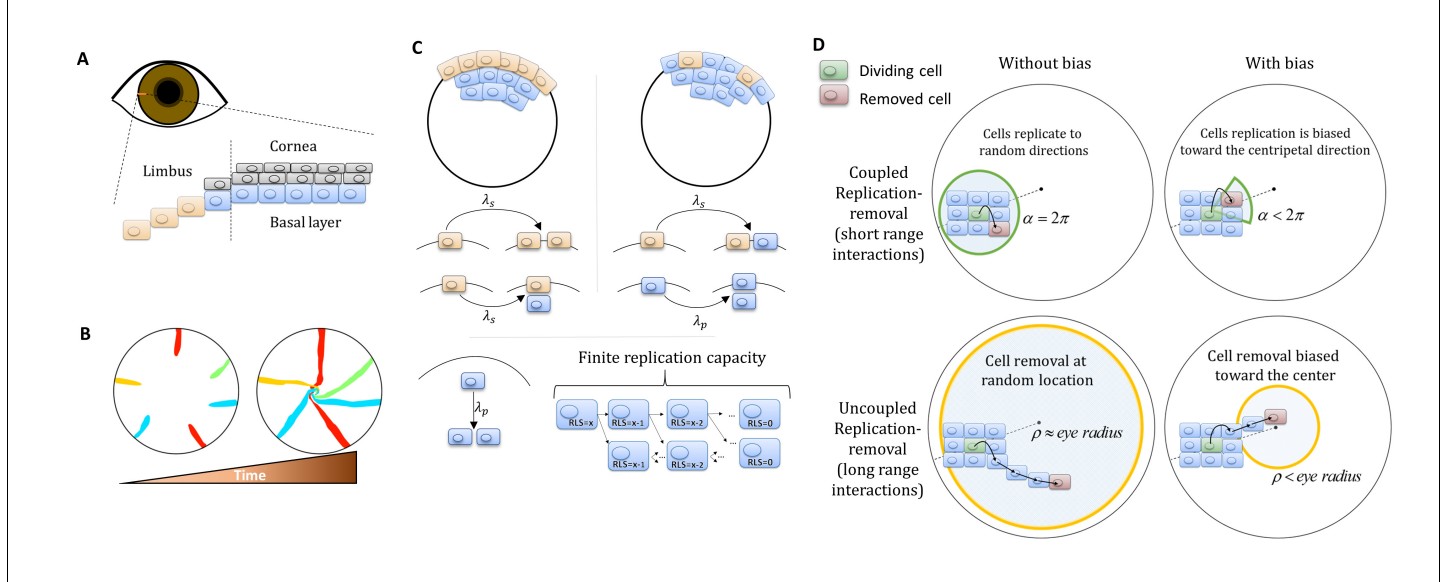

**Figure 1.** Model setup. (**A**) Maintenance of cells in the cornea, the outer transparent part of the eye, is mainly due to stem cells that reside in the limbus – a niche at the circumference of the cornea. (**B**) Illustration of in vivo multi-color lineage tracing experiment. Clonal stripes emerge from the limbus toward the center of the cornea. (**C**) We consider two limiting scenarios for stem cell distribution and dynamics. In the first (left), stem cells are evenly distributed all over the limbus. They can divide symmetrically to maintain the limbus or asymmetrically to provide progenitor cells to the cornea. In the second model (right), stem cells are scarce and divide asymmetrically to give rise to progenitor cells in the limbus that further divide and populate the cornea. In both models, progenitor cells have limited replicative lifespan capacity. When cells exhaust their replicative lifespan, the cells can no longer divide in the basal layer and are replaced by other dividing cells. (**D**) Whether the location of replication and cell removal are correlated (replication and cell removal occur in the same neighborhood) or not plays a crucial role in the cornea rejuvenation dynamics. Accounting also for whether there is a centripetal bias results in four possible classes of models.

*2016*). LESCs divide both symmetrically and asymmetrically to yield progenitor cells that have a limited replicative lifespan (RLS) (*Lehrer et al., 1998*). In turn, progenitor cells are proliferating from the limbus to the basal cornea where they can either proliferate or migrate to upper layers (*Figure 1A*; *Lehrer et al., 1998*). The observation that in homeostasis the overall number of cells in the cornea does not fluctuate dramatically led to the 'XYZ hypothesis' that states that the proliferation of epithelial cells in the limbus and their migration to the cornea is balanced by cell loss (*Thoft, 1983*). Lineage-tracing experiments in living mice revealed a pattern of clonal stripes that propagate from the limbus toward the center of the cornea (*Figure 1B* and Appendix 1 section I) (*Amitai-Lange et al., 2015*; *Di Girolamo et al., 2015*; *Dorà et al., 2015*; *Nasser et al., 2018*). Some hypotheses have been suggested regarding the mechanism of the centripetal migration dynamics in the corneal epithelium homeostasis (*West et al., 2015*), including population dynamics and electrophysiological or electrochemical cues (*Lavker et al., 1991*; *Gao et al., 2015*; *Blanco-Mezquita et al., 2013*; *Walczysko et al., 2016*; *Sharma and Coles, 1989*). Yet, the underlying mechanism behind these dynamics is not fully understood.

Using one-dimensional mathematical models to infer stem cells and progenitor cells dynamics from lineage tracing experiments have been useful in modeling different tissues such as murine epithelial homeostasis of the skin (*Mascré et al., 2012*), gut (*Snippert et al., 2010*), human epithelial homeostasis of the epidermis (*Simons, 2016*), lung (*Teixeira et al., 2013*), prostate (*Moad et al., 2017*), and the cornea (*West et al., 2018*; *Moraki et al., 2019*). Yet, while these models are insightful, the unique spatial organization in the cornea requires two-dimensional models to encompass overall pattern changes. These types of models require some assumptions on the spatial nature and organization of stem cells and on the spatial interactions between corneal cells. In particular, the role of interaction range between cells, or the spatial correlation between the location of replication and location of cell loss, plays a crucial role in the resulting dynamics (*Mesa et al., 2018*; *Miroshnikova et al., 2018*). In addition, another key property of the dynamics is the replicative lifespan (RLS) of corneal cells – the number of times each corneal cell can divide. Approximating the

tissue as a kind of elastic network that minimizes some energy function (*Barton et al., 2017*) was used to suggest that corneal epithelial cells can organize into centripetal patterns in the absence of external cues (*Lobo et al., 2016*) under the assumption that stem cells are uniformly distributed, RLS of few divisions, and no correlation between replication location and cell loss location.

In this work, we combine a novel mathematical model that allows rapid simulation of the stochastic dynamics of epithelial cells and pattern formation in the cornea together with analytical calculations to consider a broad set of possible physiological scenarios. In particular, we determine the consequences of different assumptions on the replication-removal coupling range and replicative lifespan values. We characterize the tradeoff between renewal time and replicative lifespan and determine the constraints that allow homeostasis and are consistent with the formation of the observed spatial patterns, biological timescales, and mutant dynamics.

## Materials and methods

We developed a lattice-based mathematical model of the cornea's basal layer, to examine the potential underlying mechanisms and parameters that govern corneal homeostasis, centripetal migration and spatial order patterns as seen in the in vivo data (Appendix 1 sections I and II). We model the cornea as a round assortment of cells in the basal layer of the cornea (*Figure 1C*) with a radius $R$. We assume two types of cells: stem cells ($S$) and progenitor cells ($P$) that have different doubling rates $\lambda_s$ and $\lambda_p$, respectively. $S$ cells reside only in the limbus and can either divide symmetrically or asymmetrically to produce $S$ and $P$ cells (*Figure 1C*). In the case of the 'Equipotent' model, the $P$ cells can reside only in the cornea while in the 'Hierarchical' model they can reside both in the limbus and the cornea, $P_L$ and $P_C$, respectively. $P$ cells can divide only symmetrically; if they are in the cornea they can divide in any direction and if they are in the limbus ('Hierarchical' model), they can divide only toward the cornea (*Figure 1D*). $P$ cells are also limited in their maximal amount of horizontal replications - their replicative lifespan, RLS, a parameter which will play a major role in the forthcoming results (*Figure 1C*).

Assuming corneal homeostasis, as cells in the cornea and the limbus are dividing, new cells are replacing other cells in the basal layer concurrently. The specific parameters of the replication rate, replicative lifespan, replication direction, and the coupling between the cells affect the spatial dynamics. There are two key properties that play a crucial role in tissue homeostasis dynamics, not only in the cornea. The first is the effective interaction range between replication and removal events. In one limit, the location of the cell that is removed is independent of the replication event. In the other limit, the probability of a cell to be removed will be higher in the area near the new cell (e.g., local pressure). The second property is whether there is an external bias that affects the replication direction or removal location due to, for example, chemical cues, matrix structures or local mechanical perturbation such as blinking. While previous modeling efforts focus only on a particular model, which fits the hypothesis of the study, in this work we systematically account for all these scenarios and provide the physical limitations, biological implications, and feasibility of each model.

Dynamics are simulated using a stochastic 2D lattice Monte-Carlo approach (Appendix 1 section II). In homeostasis, as cells replicate, other cells are removed to keep the overall number of cell constant. In each step of the simulation, a pair of cells is chosen: one for replication and one for removal. The locations of these cells is determined by the spatial correlation between replication and removal. In the case that replication and removal are spatially correlated, both cells are more likely to be closer to each other. To account for bias, the location of the removed cell is randomly drawn from a uniform distribution of a circular section that is facing the center of the cornea around the replicating cell, and has an angle of $\alpha$ and radius of few cells (Appendix 1 section II, *Figure 1D*). If $\alpha = 2\pi$, there is no centripetal bias, and as $\alpha$ is smaller the bias is larger (*Figure 1D*). In the case there is no correlation between replication and removal, the cells of the pair are chosen independently. To account for bias in this case, the removed cell is selected from a circle at the center of the eye with radius $\rho$. As $\rho$ is smaller, the centripetal bias increases (*Figure 1D*). Once the location of the pair, the replicated and removed cell, has been determined, the cells in the cornea reorient their location accordingly. The probability that a cell will move into the vacant hole depends on its distance from the vector that connects the replicated cell (that causes local stress) and the removed cell (which leaves a vacant space) (Appendix 1 section II). At the start of each realization, we label only the limbal stem cells and track the lineage dynamics by labeling their progenies with the same marker.

## Results

### Spatial coupling between cell replication and cell removal

First, we examine the case in which replication and removal processes are spatially coupled in the absence of centripetal bias. We will introduce the centripetal bias in the next section. In this case, the replicated and removed cells are from the same neighborhood which is an order of a few cells and is denoted by $m$ (*Figure 1D*). Accordingly, the effective step size of clone progression is small, an order of a few cells, hence there is a limit on the maximal distance of cell renewal from the limbus. This distance depends on the RLS. If the RLS is small then cells will exhaust their replicative capacity before reaching the center of the eye (*Figure 2A*, *Video 1*). For example, if the RLS is one (i.e. a $P$ cell can divide once and then becomes post-mitotic), in steady-state the renewed cell front will propagate to fill the local interaction neighborhood, $m$ (Appendix 1 section V). As RLS increases, cells can replicate further, and the renewed cell front at steady-state will be closer to the center. The steady-state location of the front is closer to the limbus than what is expected if it propagates an additional one cell toward the center for each replicative lifespan added (the deterministic limit, Appendix 1 section V). For example, even if the RLS is equal to the radius of the tissue, $R = 100$ in our case, the front still does not reach the center (*Figure 2B*), because $P$ cells lose their proliferative potential already in the periphery.

We capture the quantitative details of this phenomenon by considering an effective one-step binomial stochastic process. At each time point, the front can either stay at the same place or move forward with some probability. The probability of moving forward in the case there is no centripetal bias depends on the radial geometry of the front and is estimated to be about 3/8 (Appendix 1 section V). The results of this model are consistent with the simulation results. In the case there is no centripetal bias the expected minimal RLS, which permits replacement of central corneal cells, $RLS_{min}$, is around 130 replications (*Figure 2B*, *Video 1*). For RLSs that are above this critical value, the front can reach the center and a patched pattern is formed. The emerging pattern, in this case,

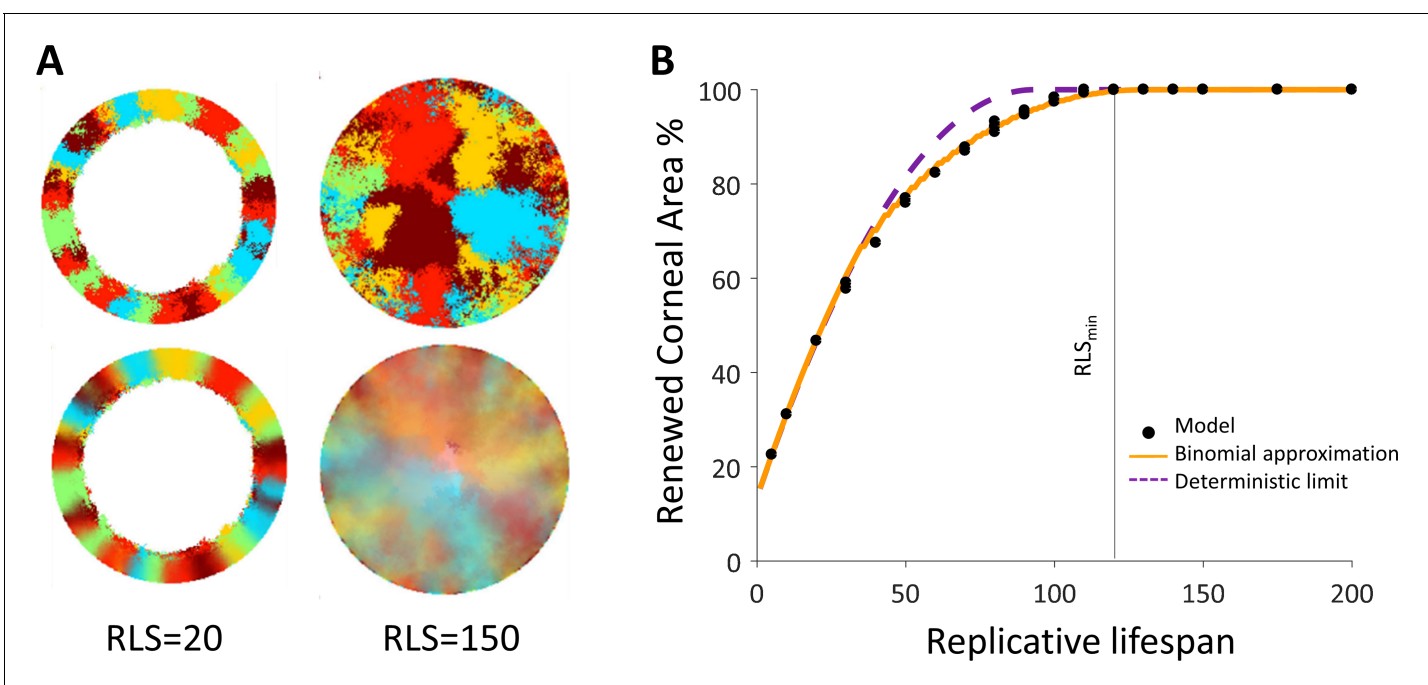

**Figure 2.** Spatial coupling between cell replication and cell removal in the absence of centripetal bias. (**A**) Steady-state snapshots (upper row) and time-averages over 200 corneal replications (lower row). If the replicative lifespan is below a critical value, the cornea cannot be rejuvenated. For replicative lifespan values that are above the critical value ($RLS_{min} \geq 130$), the emerging pattern is that of contiguous patches reminiscent of in vivo mutant phenotypes. (**B**) The fraction of the cornea that is renewed increases with replicative lifespan. Three realizations of the dynamics are shown (black dots). The purple line is the theoretical upper limit on the renewed area, and the orange line is the result of a theoretical model that approximates renewal as a one-step process (see text and Appendix 1 section V for details).

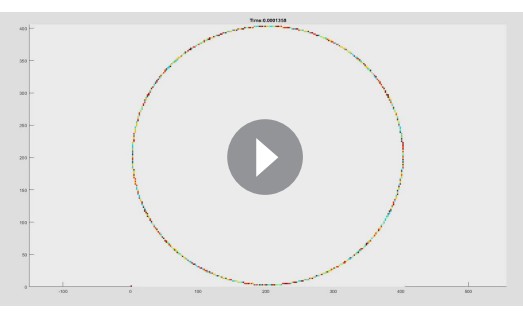

**Video 1.** Coupled spatial correlations in the absence of bias, *RLS* = 130. In this case, RLS ≥ RLS$_{min}$ and thus the cornea is fully rejuvenated. The emerging pattern is that of contiguous patches.

https://elifesciences.org/articles/56404#video1

is that of patches (*Figure 2B*, *Appendix 1—figure 2*). While this case does not provide clonal stripes, it does result in contiguous patches that resemble the in vivo pattern of mutants that lack certain genes that are thought to play a role in centripetal chemotaxis (*Findlay et al., 2016*; *Kucerova et al., 2012*; *Mort et al., 2011*; *Douvaras et al., 2013*) (Appendix 1 section III).

## Adding bias to the dynamics of coupled replication-removal

We hypothesized that biased cell division orientation toward the center (*Figure 1D*) would lower RLS$_{min}$ and lead to a pattern that resembles the in vivo clonal stripes. To capture the effect of external centripetal bias, (that could be the result of, for example, chemical or mechanical cues), once a cell replicates, the removed cell is drawn randomly from a section that is facing the center of the tissue centered around the replicated cell and has an angle $\alpha$ (*Figure 1D*). It is convenient to define the centripetal bias in this case to be between zero and one, *bias* = 1-$\alpha$/2$\pi$. As the bias increases, a centripetal pattern emerges (*Figure 3A*, *Video 2*), and the required minimal RLS for renewal goes down (*Figure 3B*). To quantify how much the pattern resembles stripes that elongate from the limbus to the center, we define a clonal unmixing parameter that captures the centripetal stripe mixing (Appendix 1 section IV). When the unmixing parameter is equal to one, the pattern is composed of perfect stripes. As the stripe order is lower, the unmixing parameter approaches zero (Appendix 1 section IV).

When there is no bias, the minimal RLS is of the order of the radius of the tissue. In the case of ideal bias, the minimal replicative lifespan required for full renewal, where *m* is much smaller than *R*, is *R/m* (Appendix 1 section V). In our case, *m* is five cells which give a minimal RLS of about 20 replications (Fig, 3B, 3B-inlet). This result is interesting in light of previous literature that attributed very limited replication capacity (RLS around 3–4) to short-lived *P* cells (*Lobo et al., 2016*; *Richardson et al., 2017*; *Lehrer et al., 1998*). It imposes a minimal lifespan for progenitor cells that scales as the radius of the tissue divided by the radius of the local neighborhood in which cells interact.

The replicative lifespan also plays a role in determining the normalized renewal time that is defined as the time it takes for all cells in the cornea to be replaced divided by the doubling time of corneal cells. As the RLS increases, the renewal time decreases (*Figure 3C*). As the centripetal bias is larger the renewal time is faster. In the case of ideal bias, the limit on the renewal time dynamics can be captured as a one-step process with an average step size that depends on the interaction length, *m* (Appendix 1 section V). The limit on the renewal time is given by 2*R*/(*m*+1) (*Figure 3C*). In our case, where the radius is about 100 cells, it results in a renewal time of ~35 replications, which amount to ~100 days assuming a 3-day cell cycle (*Lehrer et al., 1998*; *Sagga et al., 2018*).

## Spatial uncoupling of replication-removal

Spatial coupling between cell replication and cell removal has been demonstrated in skin cells, yet another possibility is the case where replication and removal are not tightly correlated in space. In this case, the replication and removal do not have to be in the same neighborhood (*Figure 1D*). First, we consider the case where

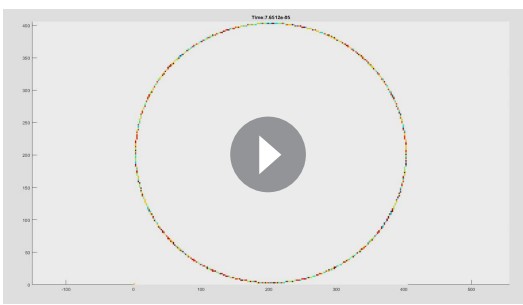

**Video 2.** Coupled spatial correlations with bias, *RLS* = 60. Adding local centripetal bias results in a pattern of centripetal stripes.

https://elifesciences.org/articles/56404#video2

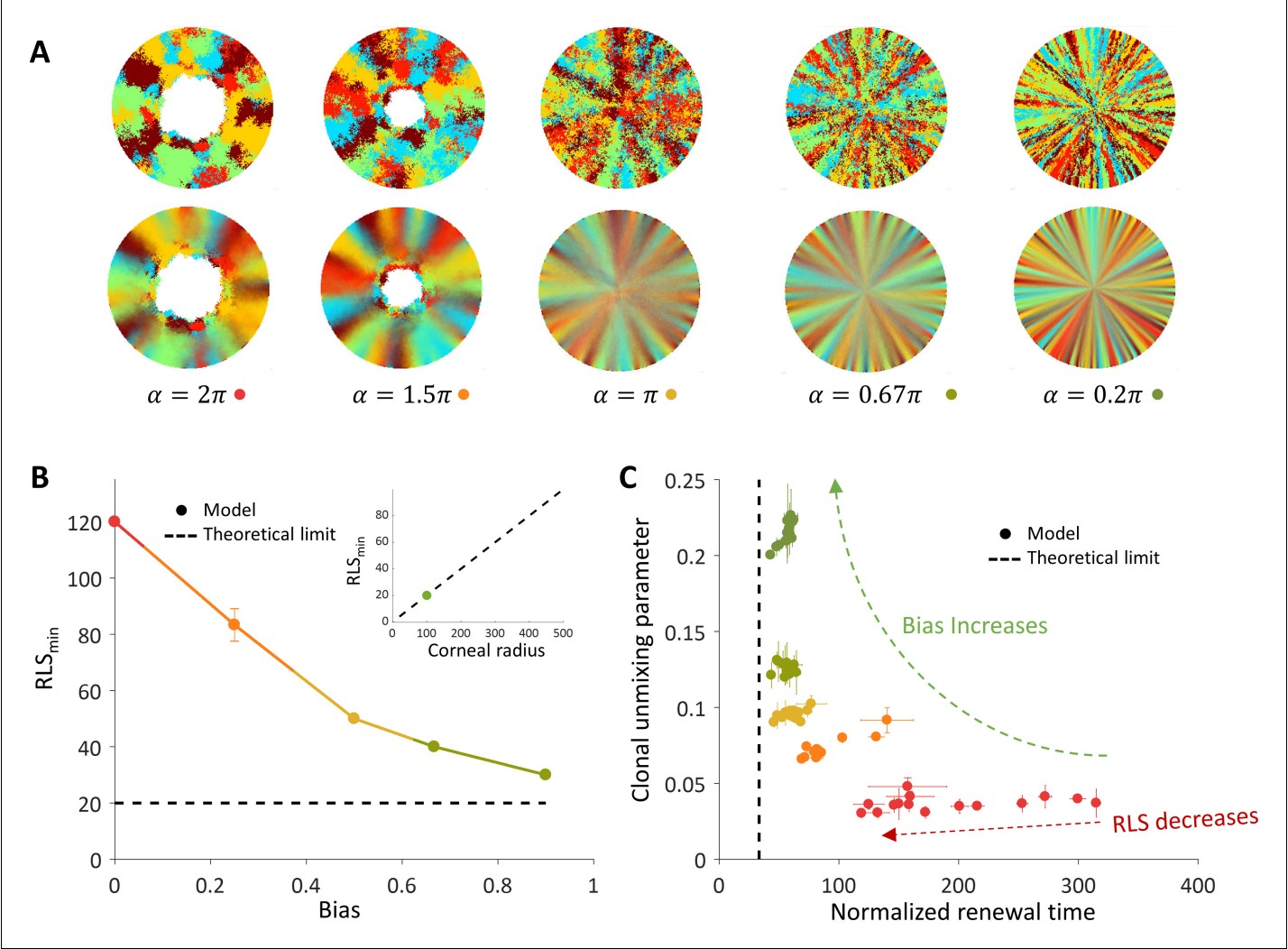

**Figure 3.** The effect of centripetal bias in the case of spatial coupling between cell replication and cell removal. (**A**) Steady-state snapshots (upper row) and time-averages over 200 replications (lower row) for $RLS$ = 60 and different centripetal bias. (**B**) The minimal RLS required for renewal as a function of centripetal bias. Data points are the mean of three realizations, and error bars (some smaller than the marker size) are the standard deviation. In the case of ideal bias, the minimal theoretical value for $RLS_{min}$ (dotted horizontal line) is ≈ $R/m$ (where $R$ is the radius of the cornea, and m is the local replication-removal neighborhood). (**B-inset**) The lower limit on $RLS_{min}$ as a function of the corneal radius when $m$ = 5. The green dot marks the corneal radius used for the simulations, $R$ = 100. (**C**) The effect of centripetal bias (Different colors, match the colors on panel 3B) on the normalized renewal time and clonal unmixing. As the bias increases, the corneal renewal time is shorter, and the pattern is more ordered. In the case of ideal bias, the theoretical lower limit on renewal time (vertical dotted line) is $2R/(m+1)$.

there is no inherent centripetal bias, that is, cell removal and replication events in the cornea are not biased toward the center. It was previously suggested that centripetal patterns can be formed even in the absence of centripetal bias (*Lobo et al., 2016*). We show here that this phenomenon is limited to a particular set of RLS and interaction lengths that are associated with slow corneal replenishment time and high post-mitotic rate.

The emergent pattern, in this case, is inherently different. For low RLS values, a centripetal pattern is formed near the limbus edge. However, the unmixing is diminishing as the stripes are moving toward the center (*Figure 4A and B*, *Video 3*), similar to previous reports (*Lobo et al., 2016*). The unmixing is decaying toward the center of the tissue and the pattern becomes mixed akin of a 'salt and pepper' noise. In this case, the disordered pattern does not form spatial neighborhoods as the coupled interaction case.

The emergence of a centripetal pattern occurs only for small RLS values. As the replicative lifespan is larger than a few replications, the emergent unmixing becomes more and more limited to

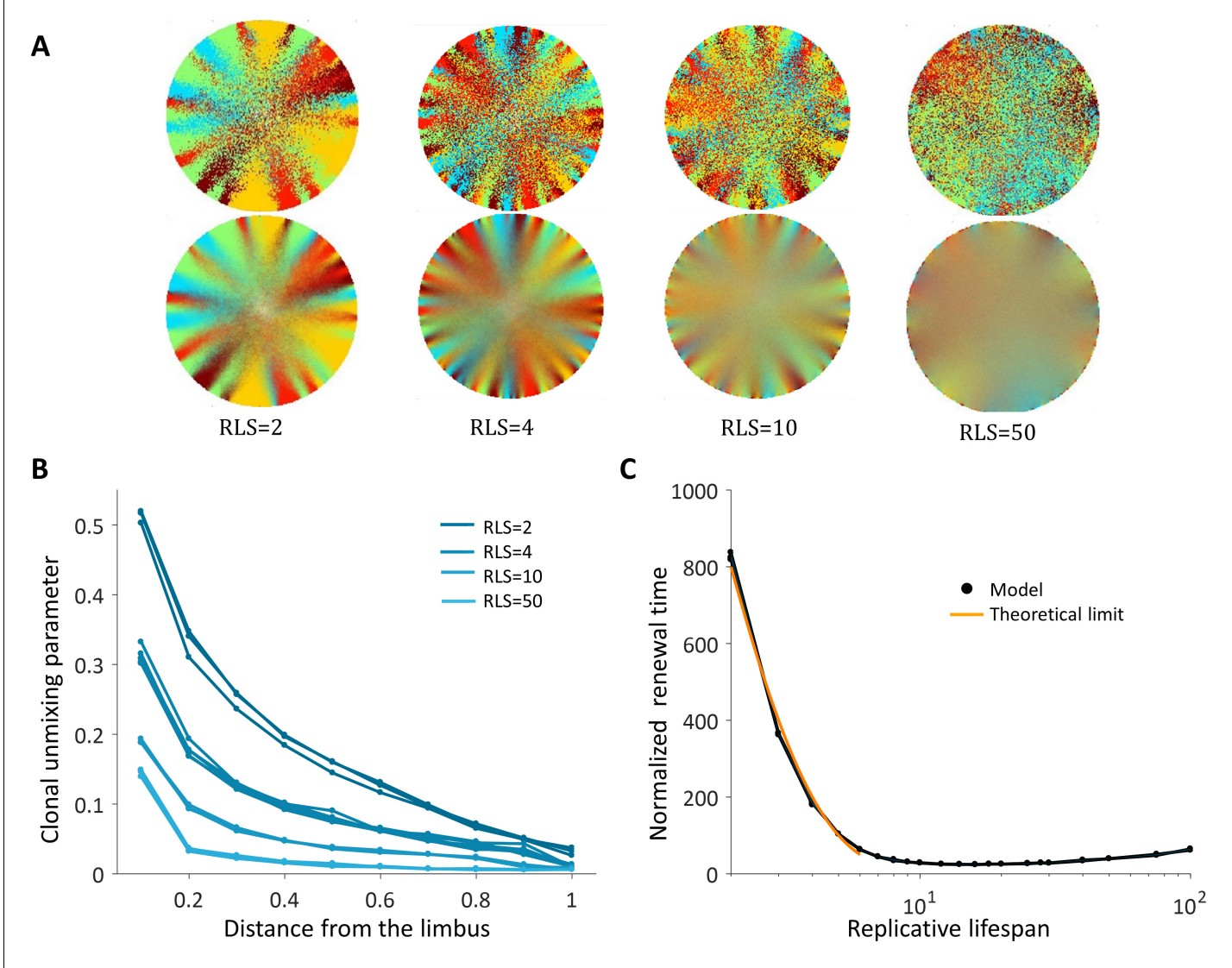

**Figure 4.** Spatial uncoupling between cell replication and cell removal. (**A**) Steady-state snapshots (upper row) and time-averages over 200 corneal replications (lower row) for different RLS values. (**B**) The clonal unmixing at different distances from the limbus. Ordered centripetal pattern emerges mainly at the periphery of the tissue and only for low values of replicative lifespan (blue-lines, three realizations for each RLS value). (**C**) The renewal time decreases exponentially with RLS for values that are lower than $\log_2(R) \approx 7$ (black line, three realizations are shown). The orange line is the theoretical limit when considering renewal as a one-step stochastic process with radial boundary (see text and Appendix 1 section VI).

the edges of the tissue. This is consistent with previous studies that assumed RLS of only a few replications. For RLS that is larger than 10, unmixed order formation is very limited. In contrast, small RLS values result in a high post-mitotic rate (Figure 7B). Interestingly, while only very low RLS values allowed some organized stripe pattern, in these low RLS values, the vast majority of the cornea was occupied by post-mitotic *P* cells that continuously underwent centripetal movement as a consequence of divisions of limbal *S* and corneal peripheral *P* cells. Thus, this regime suggests no cell proliferation at the center of the cornea.

Another ramification of short replicative lifespan is the number of replications needed to get to the center of the tissue (replication times) that also depends greatly on the RLS. When the capacity of the cells to replicate is low, RLS of only a few replications, the time that is required to get to the center in units of replication times is in the order of hundreds (*Figure 4C*). Thus, there is an inherent tradeoff between the renewal time and the centripetal order.

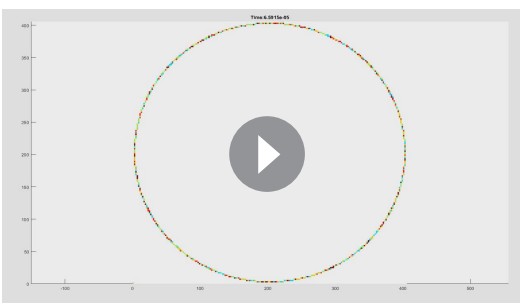

**Video 3.** Uncoupled spatial correlations with the absence of bias, *RLS* = 10. A centripetal pattern is formed near the limbus edge, and the order is diminished in the central cornea.

https://elifesciences.org/articles/56404#video3

It is insightful to estimate the limits on the maximal and minimal renewal times in this case. For the maximal time: in case the RLS is zero, that is only the stem cells can replicate, the fastest renewal time is when there is a perfect bias toward the center. In this case, the renewal time is determined by the radius of the tissue, the asymmetric replication rate of the stem cells and the geometrical difference between the number of cells that are replicating to the number of cells that are pushed forward (Appendix 1 section VI). In terms of the number of corneal cells replications, $t_{with\ bias} = \left(\lambda_p/(2\lambda_s \cdot a)\right)(R+1)$, which is about 600 corneal replication times in our case. In the case there is no bias at all, the probability to move toward the center is not one and depends on the location of the front (Appendix 1 section VI). In this case, the renewal time is larger, $t_{no\ bias} = \left(\lambda_p/(\lambda_s \cdot a)\right) \cdot H_R \cdot R$, where $H_R$ is the $R^{th}$ harmonic number. In the case the radius is 100 cells, the replenish time increase by a factor of ~10 to ~6000 corneal replication times. As the replicative lifespan increases, additional corneal cells contribute to the propagation of the front. Thus, for small RLS the replenishing time is expected to decrease by the factor $2^{RLS}$, the number of cells that contribute to pushing front. This is consistent with the results of the simulated dynamics (*Figure 4C*). For the minimal time: In case of ideal bias and high RLS, in each corneal replication time the traced stripes double their length. Thus, the minimal time needed to renew the whole cornea is $\log_2(R)$ which in our case is around 7.

## Adding centripetal bias to the uncoupled model

In the previous section, we show that if cell replication and removal are not coupled in space, the emergent clonal stripe pattern is limited to the periphery and also requires very low RLS values that lead to post-mitosis of most P cells at the corneal periphery. For these values of low RLS, the renewal time is slow and requires hundreds of corneal replications. To study the effect of centripetal bias in this case, we keep the assumption that the location of replication and removal are independent, but the location of the removed cell is from a circle that is centered at the center and has a smaller radius (*Figure 1D*). This could result from, for example, localized high pressure in the center of the eye, or from blinking that affects more the cells in the center of the eye. We note that the area with high probability for cell removal does not have to be a circle, for example, blinking can cause a horizontal, elliptic, area of high removal probability (*Ren and Wilson, 1996*; *Ren and Wilson, 1997*; *Yamamoto et al., 2002*). Here, we assume a circle to capture the qualitative tradeoffs of increasing the bias on the dynamics.

As the bias increases, the overall clonal unmixing increases (*Figure 5A and B*, *Video 4*). Yet, the overall trend of mixing order in the central region from which cells are removed is bias independent. RLS smaller than ~5 provide high unmixing but results in slow renewal dynamics (*Figure 5B*), and still high post-mitotic rate (Figure 6B). The case of uncoupled replication-removal without bias can be approximated as a 1D model. In such a model, only cell numbers as a function of time are considered and there are no spatial limitations (*Moraki et al., 2019*). Taking physiological parameters, this model yields homeostasis for RLS values of 4–12, which are comparable to the RLS range that provides minimal renewal time without bias (*Figure 5C*). It is insightful to consider two types of timescales: one is the time it takes for stripes to reach the center of the tissue, that is an important experimental observable, and the second is the overall renewal time which is the time that takes to fill the entire cornea (*Figure 5B*). These two timescales exhibit different dependencies on the replicative lifespan. As the RLS is larger, the time it takes a clone stripe to reach the center is increasing exponentially with RLS. For large values of RLS, as the bias is larger, the velocity of stripe progression is larger. The limit on stripe speed can be estimated in the case of ideal bias, that is the removed cells are from the center of the tissue, and high replicative capacity, the minimal number of

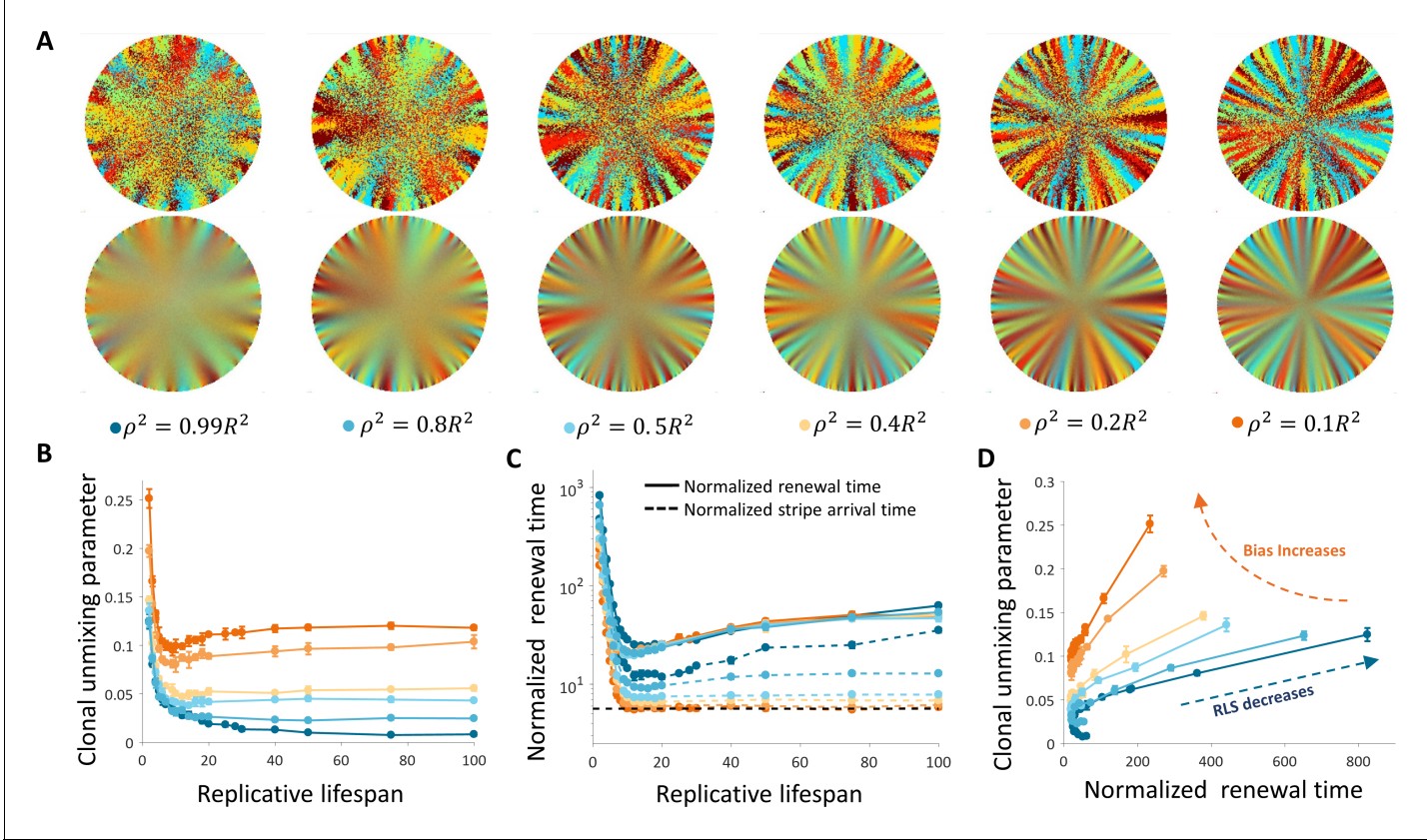

**Figure 5.** Dynamics and patterns in the case where replication and removal are uncoupled. (A) Steady-state snapshots (upper row) and time-averages (lower row) over 200 corneal replications for different values of $\rho$, the radius of the area from which cells can be removed, for $RLS = 10$. (B) The dependence of centripetal unmixing, halfway to the center, on replicative lifespan. Colors denote the centripetal bias and are the same as in 5A. Data points are means of three realizations, and error bars are the standard deviation. For all values of RLS, the order decreases as RLS increases. (C) Normalized renewal time decreases as RLS increases (solid lines). The black dotted line is the theoretical limit in the case of ideal bias and high RLS. Data points are means of three realizations, and error bars are standard deviation. (D) The interplay between unmixing and renewal time for different bias (different colors) and different RLS values ranging from 2 to 100. Data points are means of three realizations, and error bars are standard deviation.

replication needed for a stripe to reach the center is given by $\log_2(R)$. (Appendix 1 section VI), where $R$ is the radius of the tissue, which is around seven corneal replication times in the case $R = 100$. The renewal time of the entire tissue dynamics exhibits a non-monotonic behavior. For high RLS values, the centripetal motion slows down the motion in the direction which is orthogonal to the centripetal direction and thus slows down the overall renewal dynamics. The interplay between the renewal time and unmixing is shown in *Figure 5C*.

## Limits on the different models

We have considered three main properties that affect the dynamics and pattern formation in the cornea: spatial correlation between replication and removal, inherent centripetal bias, and replicative lifespan of cells in the cornea. For different models, there were different constraints that determine their biological feasibility. Under conditions of spatial coupling between cell

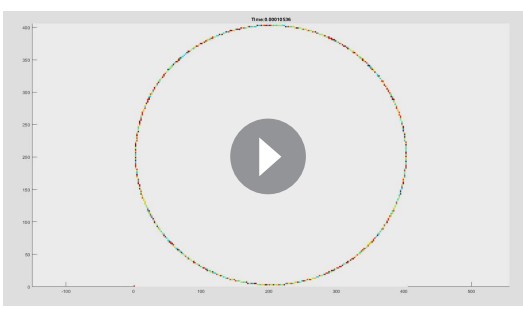

**Video 4.** Uncoupled spatial correlations with bias, *RLS* = 10. Adding centripetal bias increases the overall clonal unmixing. Yet, the central region is less ordered. https://elifesciences.org/articles/56404#video4

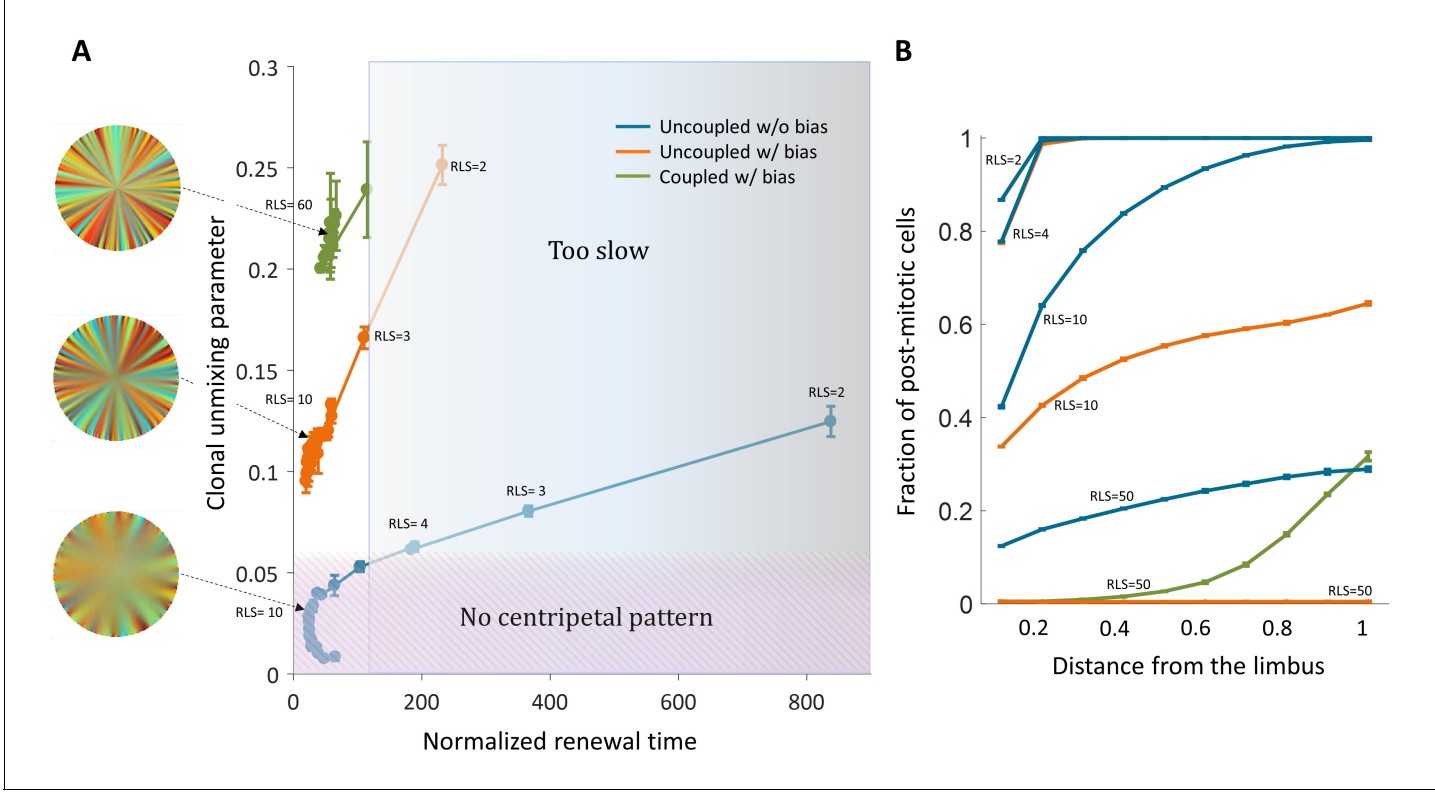

**Figure 6.** Limits and constraints on the different model classes. (**A**) The unmixing parameter as a function of the normalized renewal time for different values of replication lifespan, ranging from 2 to 100. The shaded areas are regions that are not consistent with experimental observations of centripetal unmixing and renewal time. (**B**) The hallmark of low RLS models is a high post-mitotic fraction regardless of bias.

replication and cell loss, external cue that imposes a bias of centripetal movement is required for the emergence of a centripetal pattern. Another major constraint is a requirement for a minimal value of RLS that allows renewal. This value depends on the replication bias and cannot be shorter than a life-span of ~20 replications. Once the condition for the minimal RLS is met, dynamics are feasible, and the emerged unmixing is higher than the unmixing of long-range interactions.

Recent nucleotide analogue incorporation experiments suggest a progenitor cell cycle of 3–5 days (*Sagga et al., 2018*), and complete renewal of the cornea occurs, in mice, in few months (*Amitai-Lange et al., 2015*). Thus, the normalized renewal time is in the order of 20–50 replication times, setting 100 replication times as a conservative upper limit (*Figure 6A*). In the case of uncoupled interactions without bias, centripetal unmixing formation is feasible under very specific constraints. Centripetal clonal unmixing emerges but mainly in the periphery of the tissue and only for small RLS values that are below ~10. However, for these RLS values, the renewal dynamics and stripe propagation velocity, of a few hundred corneal replications are much slower than experimental observations (*Figure 6A*). Adding bias to these types of interactions allows formation in feasible time scales and the emergent of a centripetal pattern for slightly lower RLS values (*Figure 6A*).

Another experimental observation is the fact that in some mutants that abolish centripetal bias, the resulting pattern is that of contiguous patches. Models of uncoupled interaction cannot provide such a pattern while coupled can. In the case of uncoupled interactions without bias, the resulting pattern would be akin to 'salt and pepper' mixed pattern rather that of patches. Thus, only the coupled model can explain contagious patches in mutants with the constraint of RLS >20.

Another outcome of low replicative lifespan is the distribution of post-mitotic cells. *Figure 6B* shows the fraction of post-mitotic cells as a function of the distance from the limbus. Models that require a low replicative lifespan, such as uncoupled replication-removal without bias, result in a complete post-mitosis of the central cornea. A scenario of post-mitotic central cornea is not consistent with in vivo data; nucleotide analogue labeling experiments suggest that some

(*Richardson et al., 2017*) or most of the cells in the central cornea are actively mitotic (*Sagga et al., 2018*).

## The effect of stem cells distribution and dynamics

There are two limiting common hypotheses for the properties of stem cells, and in particular limbal stem cells, in their niche. In the previous section, we investigated the case where stem cells populate all the limbal cells and can replace each other as they replicate. In this section, we also consider the case in which stem cells are rare cells (~10%) (*Sartaj et al., 2017*) which divide asymmetrically to limbal progenitor cells that in turn, divide into corneal cells. In this case, the stem cells cannot be replaced by limbal progenitor cells. The interplay between the emergent pattern and the

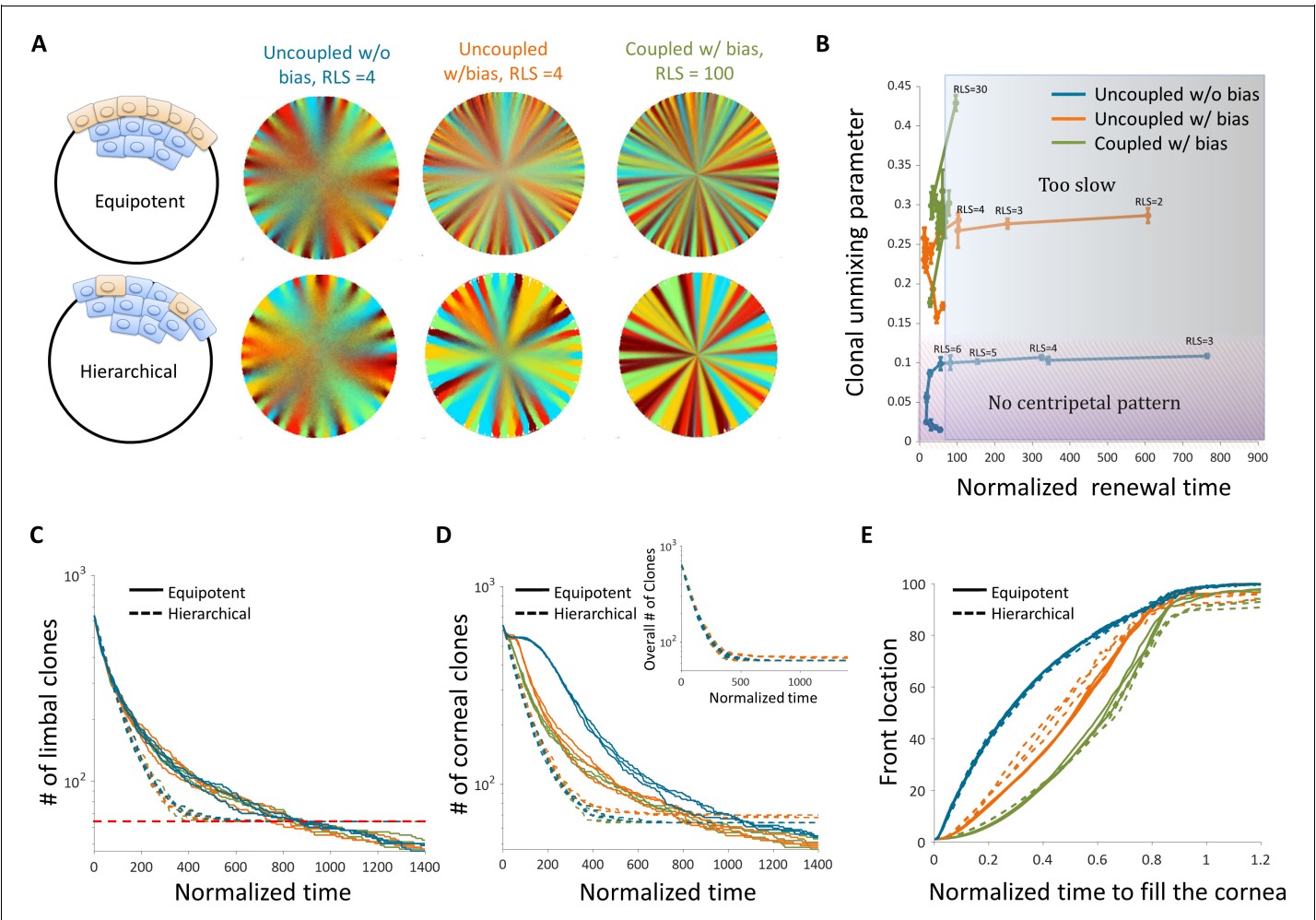

**Figure 7.** The effect of stem cell dynamics and distribution. (**A**) Steady-state snapshots (upper row) and time-averages (lower row) over 200 corneal replications for the Equipotent and Hierarchical models. The colors denote the different cases (blue, orange, and green) and are the same for all the panels. (**B**) The interplay between clonal unmixing and renewal time for different bias and different RLS values in the case of the Hierarchical model. The tradeoffs are similar to those of the Equipotent model (*Figure 6A*). (**C**) The number of limbal clones as a function of time in the case of the Equipotent model (solid lines) and the Hierarchical model (dotted lines). While the number of limbal clones in the equipotent case diminishes with time, the number of limbal clones in the hierarchical case approaches the number of limbal stem cells (red horizontal dotted line). In both models, the spatial coupling does not affect the dynamics of clone number. (**D**) The number of corneal clones as a function of time in the case of the Equipotent model (solid lines) and the Hierarchical model (dotted lines). In the hierarchical case, the dynamics of corneal clone number and limbal clone number are similar (inset). In the equipotent case, the spatial coupling affects the decay rate of the number of clones. (**E**) The renewed cells' front location as a function scaled time. Time was scaled such that 1 is the time to replenish the cornea. The stripe propagation velocity depends on the spatial coupling but is less sensitive to the stem cells distribution.

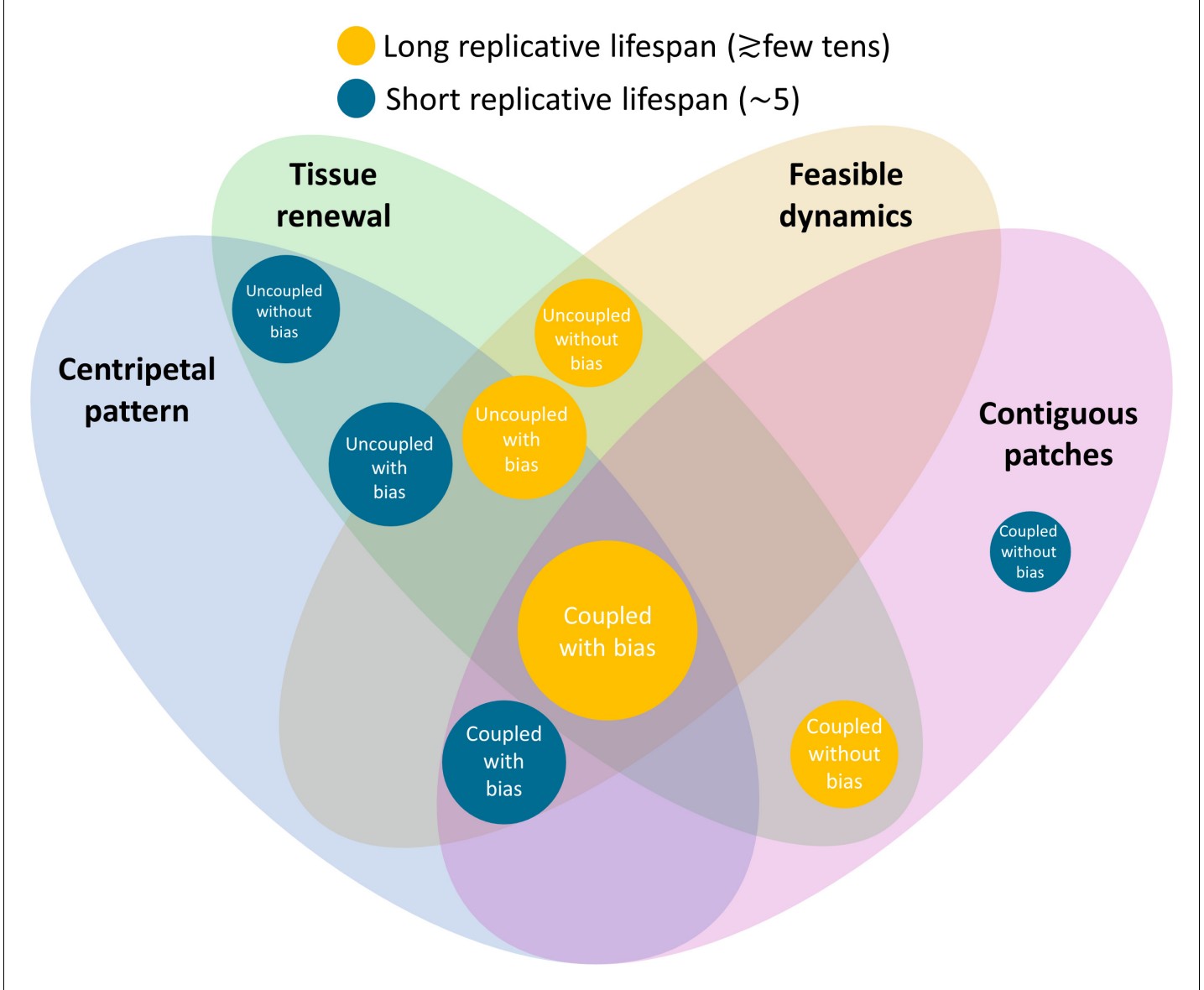

**Figure 8.** Constraints on the different models and their biological feasibility. Each circle represents a model with different spatial correlations between cell replication and cell removal, high or low replicative lifespan, and whether there is a centripetal bias or not. The colors represent whether the replicative lifespan is short or long (blue and orange, respectively). The radius of each circle is proportional to the number of properties each model is consistent with. In the case of the cornea, the only model that can provide all four requirements (centripetal pattern, full tissue renewal, feasible time scales, and contiguous patches observed in mutants) is coupled replication-removal with bias and long replicative lifespan.

dependence on replicative lifespan and replication-removal interaction length is similar to the case of uniform progenitor cells (*Figure 7A and B*).

To highlight the differences between these two models, it is insightful to examine the number of clones as a function of time both in the limbus and in the cornea. In the case of the Hierarchical model, both the number of clones in the limbus and in the cornea goes down in a similar fashion until it is equal to the number of stem cells. (*Figure 7C*). These dynamics are invariant to the interaction length and to the degree of centripetal bias. In the case of the Equipotent model, the dynamics of the number of clones in the limbus and in the cornea is different. In the limbus, there is a monotonic decline in the number of clones. This decline is invariant to the interaction length due to the continuous competition in the limbus. It is insightful to compare the Equipotent model to the neutral drift clonal competition model in which the main predictions are that the number of clones declines

as $1/\sqrt{t}$, the average clone size increases as $\sqrt{t}$, and the distribution of clone size divided by the average ($n/<n>$) is time-invariant (*Klein and Simons, 2011*).

In the case of the Equipotent model, the dynamics of the limbal clone number and average limbal clone size are monotonically increasing and monotonically decreasing, respectively. Yet, they do not follow the neutral drift dynamics during the entire time trajectory (Appendix 1 section VIII, *Appendix 1—figure 5*). However, the limbal clone size distribution does exhibit scale invariance (*Appendix 1—figure 5*). The dynamics of the number of clones in the cornea has a bi-phasic shape that does depend on the interaction length (*Figure 7D*). The first part of the bi-phasic dynamics is a decline that is similar to the limbus decline and is due to stem cell competition. A plateau follows this decline as clones are propagating toward the center. The second decline is due to the clonal competition when all the tissue is labeled.

Another experimental observable is the location of the clonal front. The normalized front propagation velocity is invariant to the stem cell dynamics (whether it is the Hierarchical or Equipotent model) and depends mainly on the interaction length and bias (*Figure 7E*). Moreover, the front location as a function of normalized time, suggests that the velocity in which the front propagates depends on the radial location. A concave curve indicates that the front velocity accelerates as it is closer to the center, while a convex curve suggests a slowdown (Appendix 1 section VII, *Appendix 1—figure 4*). In the case there is a centripetal bias, the velocity of the front propagation is accelerating as they are moving away from the stem cell niche toward the center (*Figure 7E*, *Appendix 1—figure 4*). This is similar to the dynamics in the intestinal crypts, where cells that are higher on the crypt axis move faster (*Tóth et al., 2017*).

## Discussion

The dynamics of corneal stem cells and their progenitors play a key role in maintaining homeostasis in adult tissues. As the total number of cells in homeostasis remains constant, the main facilitators of cell location and tissue rejuvenation, when the tissue is intact, are cell division and removal. Thus, the spatial correlation between the locations of the replicated cell and the removed cell determines the rejuvenation speed and clonal pattern of the tissue. These spatial correlations can arise from mechanical or chemical interactions. The actual mechanical interaction is complex as it is affected by many factors such as interactions with the matrix, interactions between cells, and the cornea's geometry. Moreover, these mechanical interactions can be long-ranged (e.g., in the elastic limit) or dictated by local interactions. In this work, we accounted for two effective spatial correlations between the location of the replicated and removed cells: a short-range interaction with a typical interaction range of $m$ cells (the 'coupled' model) and the limit of long-range interactions (the 'uncoupled' model).

Another feature that is critical in replication-removal dynamics is the replicative lifespan of the progenitor cells. Here, we use a mathematical model together with analytical benchmarks to derive the tradeoffs and constraints of varying replication-removal correlation length and replicative lifespan and characterize the conditions that are consistent with experimental measurements. Identification of the conditions governing corneal cell dynamics will facilitate new approaches to limbal stem cell deficiency treatments and translate to other cellular systems that are dependent on spatial cell arrangement and division.

Spatial coupling of replication and removal dramatically influences the parameters that are needed for tissue renewal in physiological time scales. Recent studies suggested that replication and removal events in homeostasis happen in close proximity of a few cells (*Mesa et al., 2018*; *Miroshnikova et al., 2018*). The main consequence of this type of 'short-range' interaction is that they set a minimal replicative lifespan (*Figure 3B*). The limit for replicative lifespan is the ratio between the radius of the cornea and the radius of the local neighborhood in which replication and removal occur. For example, in the case of a cornea with a radius of ~100 cells and a local interaction neighborhood with a radius of ~5 cells, the minimal replicative lifespan should be at least 20 replications. This limit is in the case of high bias, that is the cells are replicating toward the center of the tissue. In practice, one should expect a higher limit (*Figure 3B*). This suggests that if cell replication and cell removal are indeed spatially correlated, the replicative lifespan of progenitor cells should be much higher than traditional values which are an order of only a few replications. Another limit is imposed by the replication-removal interaction range, $m$. We showed that $m$ dictates $RLS_{min}$

(Appendix 1 section V), and tissue renewal time (*Figure 3C*). As the interaction range is smaller, the minimal RLS needed for a complete renewal of the cornea is higher and scale as $R/m$. For interaction range that is of the order of the cornea radius, the minimal RLS scale as $\log_2(m)$ (Appendix 1 section V). That is, mechanical or chemical interactions that are local require a higher replication capacity of corneal cells. The relation between the interaction range and cornea renewal time is monotonic, $t = 2R/(m+1)$ (Appendix 1 section V). Future work can account for a more complex spatial structure of the correlations that capture more intricate mechanical interactions between the cells and the tissue.

Another interesting consequence of these results is that increasing the tissue size (e.g. human and large mammals cornea and large organs) requires increasing the replicative lifespan of progenitor cells or increasing the local replication-removal interaction length. Cancer, aging or other hyperplastic conditions (e.g. psoriasis) are extreme examples of potentially extensive changes in replicative lifespan that may lead to failure in maintaining tissue renewal and proper tissue size, leading to a burden on stem cells, failure to maintain homeostasis, and/or regenerate the tissue under stress (Appendix 1 section IX).

In this context, it is insightful to consider the limits on mutation accumulation quantitatively (*Frank and Nowak, 2004*; *Vermeulen et al., 2013*). Cancer in the cornea is relatively rare, and when it occurs, it is believed to originate from the conjunctiva or the limbus (*Basti and Macsai, 2003*). The probability of mutation accumulation that results, for example, in cancer depends on the probability of acquiring oncogenic (or other deleterious) mutations, the number of cells, and the number of replications before the mutated linage is removed. In the cornea, the number of cells is low compared to the number of cells in the skin or in the gut, for example (*Milo et al., 2009*). In general, symmetric divisions promote cell exchange and can flush out mutations, and therefore the Equipotent model is more resilient to mutations in the limbus (Appendix 1 section IX). Mutation accumulation potential of a linage that originates from a mutator in the limbus depends on the linage lifetime and depth. Centripetal bias decreases the depth of a linage, and the radius of the cornea imposes a limit on the median depth (Appendix 1 section IX, *Appendix 1—figure 6*). The typical linage depth, together with the small number of cells in the cornea, suggest a low probability for an accumulation of a series of oncogenic mutations in comparison to other organs such as the skin or the gut.

We also characterized the dynamics in the limit in which replication and removal events are not spatially correlated. The case without any spatial consideration, such as bias and replication-removal correlations, can be approximated by a 1D model. A model, which is a private case of our work, suggested that physiological homeostasis requires RLS in the range of 4–12 replications (*Moraki et al., 2019*). This estimate is consistent with our Equipotent uncoupled model in the absence of bias. A 2D uncoupled replication-removal without bias was suggested in the context of the cornea with a replicative lifespan of fewer than about five replications (*Lobo et al., 2016*). Our results show that while rejuvenation of the entire cornea is possible for short replicative lifespan, the rejuvenation time without external bias is much slower than physiological estimations (*Figure 6A*). Thus, self-organizing stripe formation in homeostasis without external cues, while possible, is very limited and results in a rejuvenation time of hundreds of replications, that is, hundreds of days assuming corneal replication time of a few days. Another hallmark of a model that has a short replicative lifespan is a cornea in which most of the cells are post-mitotic (*Figure 6B*).

*Figure 8* summarizes the predictions of each model and its consistency with experimental data. We focused on four main experimental attributes: (1) Tissue renewal: whether the model allows complete rejuvenation of all cells in homeostasis. (2) Feasible dynamics: Whether the speed of clonal spread is physiological. (3) Centripetal pattern: Whether the model allows the formation of centripetal clonal stripes. (4) Contiguous patches: Whether the model allows the formation of contiguous clonal patches that is reminiscent of VNGL/PAX6 mutants. The model that seems to account for all features is that of coupled replication-removal dynamics ('short-range interactions') with centripetal bias and a replicative lifespan that is at least ~20 replications. One of the main predictions of such a model is that cells should proliferate not only near the limbus but also closer to the center of the cornea.

Our results regarding the interplay between replication-removal interaction length and replicative lifespan do not depend on whether stem cell dynamics and distribution follow the 'Hierarchical' or 'Equipotent' model (*Figure 7B*). As expected, our results show that the number of clones overtime

on the limbus and cornea together could distinguish between the two. While in the 'Equipotent' model, the number of clones in the cornea has a certain plateau and delay between the cornea and the limbus while in the 'Hierarchical' model there is no difference in the dynamics of the number of clones in the limbus and in the cornea (*Figure 7C*; *Figure 7D*).

While stem cells, that are considered as long-lived cells that can self-renew, are at the focus of regenerative medicine, progenitor cells, that are viewed as short-lived cells with a very limited replication potential, are often overlooked. Our work highlights the crucial role of replicative lifespan of progenitor cells in shaping rejuvenation dynamics in homeostasis. Our conclusions regarding the interplay between replication-removal locality and replicative lifespan are relevant for any tissue in which conditions do not permit significant cell motility and thus spatial homeostasis is maintained through cell replication.

## Acknowledgements

We would like to thank Tanya Wasserman and Waseem Nasser for their critical input and constructive comments. This work was supported by the American Federation for Aging Research, Israel Science Foundation grants 1619/20, 1308/19, 2830/20, Rappaport Foundation, The Prince Center for Neurodegenerative Disorders of the Brain #828931, and the National Institutes of Health R21 800040.

## Additional information

### Funding

| Funder | Grant reference number | Author |
| --- | --- | --- |
| American Federation for Aging Research | New Investigators Awards in Alzheimer's Disease | Yonatan Savir |
| Israel Science Foundation | 1619/20 | Yonatan Savir |
| Rappaport Foundation | | Ruby Shalom-Feuerstein Yonatan Savir |
| Israel Science Foundation | 1308/19 | Ruby Shalom-Feuerstein |
| Israel Science Foundation | 2830/20 | Ruby Shalom-Feuerstein |
| The Prince Center for Neuro-degenerative Disorders of the Brain | 828931 | Ruby Shalom-Feuerstein Yonatan Savir |
| National Institutes of Health | R21 800040 | Ruby Shalom-Feuerstein |

The funders had no role in study design, data collection and interpretation, or the decision to submit the work for publication.

### Author contributions

Lior Strinkovsky, Conceptualization, Data curation, Software, Formal analysis, Investigation, Visualization, Methodology, Writing - original draft, Writing - review and editing; Evgeny Havkin, Software, Formal analysis, Validation, Investigation, Visualization, Methodology, Writing - original draft, Writing - review and editing; Ruby Shalom-Feuerstein, Conceptualization, Methodology, Writing - review and editing; Yonatan Savir, Conceptualization, Resources, Data curation, Software, Formal analysis, Supervision, Funding acquisition, Validation, Investigation, Visualization, Methodology, Writing - original draft, Project administration, Writing - review and editing

### Author ORCIDs

Lior Strinkovsky (iD) https://orcid.org/0000-0002-0301-3515
Yonatan Savir (iD) https://orcid.org/0000-0002-5345-8491

### Ethics

Animal experimentation: This study was performed in strict accordance with the ARVO Statement for the Use of Animals in Ophthalmic and Vision Research. All experiments were approved by the Technion ethical committee, reference #IL0980713.

### Decision letter and Author response

Decision letter https://doi.org/10.7554/eLife.56404.sa1
Author response https://doi.org/10.7554/eLife.56404.sa2

## Additional files

### Supplementary files

• Transparent reporting form

### Data availability

All data generated or analysed during this study are included in the manuscript and supporting files.

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

## Appendix 1

### I. Experimental linage tracing

Current linage tracing experimental setups are based on the confetti mouse system (such as double transgenic K14-Cre$^{ERT2}$; Brainbow$^{2.1}$ or UBC-Cre$^{ERT2}$; Brainbow$^{2.1}$) in which injection of tamoxifen results in a stochastic, irreversible, expression of fluorescent markers. This expression can occur in all the basal epithelial cells, including basal limbal epithelial cells and basal corneal epithelial cells. With time, radial stripes that emerge from the limbus can be distinguished, and their kinetics can be measured (*Figure 1B*). Moreover, with time, these stripes constitute the majority of the labeled area (*Amitai-Lange et al., 2015*). Our model aims to capture the role of cellular properties, such as replicative lifespan, and tissue properties, such as spatial correlations between cell replication and cell removal on radial stripes dynamics in homeostasis. As we focus on the radial stripes dynamics and data, the labeled cells in the model originate from the labeling of limbal stem cells.

*Appendix 1—figure 1* shows typical linage tracing radial stripes of mouse 16 weeks after tamoxifen injection side by side with a simulation.

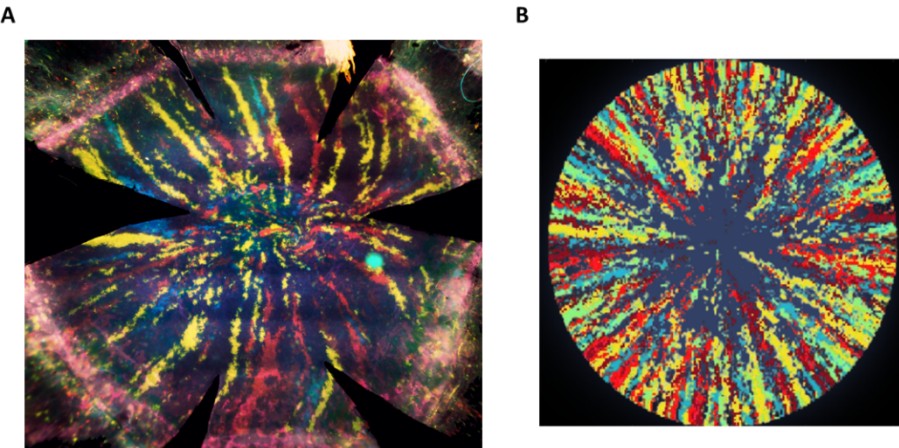

**Appendix 1—figure 1.** Radial stripes as measured in an in vivo lineage tracing experiment. (**A**) A fluorescence microscopy photomicrograph of a cornea from a double transgenic UBC-Cre$^{ERT2}$; Brainbow mice were injected with tamoxifen when six weeks old and sacrificed 16 weeks later. Tamoxifen treatment induced a stochastic and irreversible expression of one out of four "confetti" fluorescent proteins (namely, nuclear green (GFP), cytoplasmic red (RFP), cytoplasmic yellow (YFP), or membrane cyan (CFP) fluorescent protein). (**B**) Image representation of our model with the following parameters: spatial coupling of replication-removal, full centripetal bias, and replicative lifespan (RLS) of 100.

### II. Stochastic 2D lattice Model

#### Model setup

We modeled the corneal epithelium as a circle on a square 2D lattice, where each pixel is a cell. The limbus is a one-cell (that is, a one-pixel wide ring) perimeter in the circumference of the cornea. Stem cells (*S*) reside in the limbal region in the circumference of the cornea. They are also characterized by immense replicative capacity rendering them 'immortalized' for the sake of the model. Progenitor cells (*P*) that reside in the cornea are limited in their replicative lifespan (RLS): after a defined number of divisions, they cease to divide on the corneal plane.

In the case of the 'Equipotent' model (*Figure 1A*), the entire limbus is composed out of *S* cells that divide in the limbus with a rate $\lambda_s$. The cells can divide asymmetrically with probability $p_a$ and give rise to a *P* cell that resides in the cornea or divides symmetrically with a probability of (1-$p_a$), both *S* cells remain in the limbus. In the cornea, *P* cells replicate with a rate $\lambda_p$. The cells that

originate from the stem cells, have RLS replications (*Figure 1B*). That is, the maximal number of cells from each *P* cell is $2^{RLS}$.

In the case of the 'Hierarchical' model (*Figure 1A*), only a small fraction of cells in the limbus, $f_S$, is *S* cells. These cells are distributed uniformly in the limbus. The *S* cells can divide asymmetrically with a rate $\lambda_s$. The resulting progenitor cells, $P_L$, remain in the limbus and divides with a rate $\lambda_p$ to give rise to a limbal and corneal *P* cells, $P_L$ and $P_C$, respectively. Both $P_L$ and $P_C$ have finite RLS.

We used a radius of *R* = 100 cells to match the in vivo mouse data (*Appendix 1—table 1*). A distance of 50 pixels was taken from the central pixel to define the pixels of the boundary (that is, the horizontal and vertical axis contains 101 pixels). The one-pixel wide ring in the circumference is marked as the limbus. The cornea is maintained in homeostasis. Thus, cell number is constant and in each step of the simulation cell division is concurrent with cell desquamation. The dynamics of spatial rearrangement of cells in the grid are based on a 'pushing' mechanism that represents the effective mechanical interactions between the cells. In this mechanism, cells reorient in a pushing manner from the duplicated cell to the vacant hole (see details below).

## Simulation steps

We used a 2D Monte-Carlo (MC) simulation. In each MC step of the simulation, a pair of a replicating cell and a removed cell is selected as described below. *P* cells exceeding their RLS cannot be selected as a dividing cell. The probability of choosing a cell for replication depends on the replication rates according to the Gillespie algorithm (*Gillespie, 1977*; *Gillespie, 1976*; *Gibson and Bruck, 2000*).

The selection of the removed cell depends on the model class: whether the replication-removal are coupled or uncoupled (short-range or long-range interaction), and on the magnitude of the centripetal bias (see details below).

1. If the replicated cell is a corneal cell, that is a *P* cell (in the Equipotent case) or a $P_C$ cell (in the Hierarchical case) the removed cell will be in the cornea as well. The location of the removed cell will be:
    a. Coupled replication-removal: Randomly selected from a circular sector with a radius *m* and an angle $\alpha$. The center of the circle is the replicated cell, and the sector is oriented towards the center of the cornea (*Figure 1D*).
    b. Uncoupled replication-removal: Randomly selected from a circle around the center of the cornea and has a radius $\rho$ (*Figure 1D*).
    c. After the pair of a replicating cell and a removed cell is selected:
        i. The replicating cell generates a *P* cell that is in a random direction from the replicating cell and in a distance of 0.5 cell grid distance off-grid.
        ii. The algorithm finds the path of cell replacement closest to a straight line that connects the replicating cell and the 'hole' that is left on the removed cell location.
        iii. The path is based on eight directional movements on the square lattice. Excluding directions that make the path cross the limbus of the cornea or exit the boundaries of the cornea.
        iv. On the generated path, the cell closest to the hole moves into it, creating a new 'hole'. The next cell on the path that is closest to the hole moves into it and so on until the new progenitor cell which was off-grid replaces the last cell in the path.('Pushing')
2. If the replicating cell is a limbal cell (*S* in the Equipotent case; *S* or $P_L$ in the Hierarchical case):
    a. In the Equipotent case:
        i. *S* cells divide asymmetrically with probability $p_a$. In this case, the *S* offspring remains in the location of the parental *S* cell in the limbus and the *P* offspring becomes a corneal cell. The removed cell will be from the cornea and the location of the removed cell and the rearrangement of cells location is the same as in 1.
        ii. *S* cell divides symmetrically with a probability (1-$p_a$). In this case, the removed cell is chosen randomly from the limbus. The rearrangement of the limbal cells follows the same steps as in 1c, with a path that is confined to the limbus.
    b. In the Hierarchical case:
        i. *S* cells divide asymmetrically. The *S* offspring remains in the parental *S* cell in the limbus and the $P_L$ offspring remains a limbal cell. The removed cell is chosen randomly from the $P_L$ cells that are between the two sides of the parental *S* cell and it neighboring *S*

cells. The rearrangement of the limbal cells follows the same steps as in 1 c, with a path that confined to the limbus.

 ii. $P_L$ cells divide to produce a corneal $P$ cell, $Pc$. the $P_L$ offspring remains in the location of the parental $P_L$ cell in the limbus and the $P_C$ offspring becomes a corneal cell. The removed cell will be from the cornea and the location of the removed cell and the rearrangement of cells location is the same as in 1.

## Initial conditions

The simulation is initialized as follows:

1. A square lattice grid is created with a size of ($2R+3 \times 2R+3$) while $R$ equals the radius of the cornea. A circle representing the cornea is also created around the center of the grid with the radius $R$. We used $R = 100$ cells to match the in vivo data.
2. A 1 cell wide ring in the circumference is marked as the limbus.
3. Each $S$ cell in the limbus is given a distinctive lineage marker, numbered from 1 to 642 ($\sim 2\pi R$).
4. The RLS of $P$ cells is initialized to the steady-state distribution determined by running the simulation for long periods of time.
5. For visualizing the data, each $S$ cell is also color-coded randomly with a color index from 1 to 5 to create the supplementary movies.

## III. Patterns without bias

To highlight the differences in the emerging patterns, *Appendix 1—figure 2* illustrates the realizations of a few cases where there is no centripetal bias and there is only one labeling color. Half of the limbal stem cells were randomly selected for labeling. In a similar way to *Figure 2A* in the main text, in the case of coupled replication-removal, the resulting pattern is that of contiguous patches (*Appendix 1—figure 2A, C*). In the case of uncoupled replication-removal, the pattern is inherently different; there are stripes in the periphery near the boundary and 'salt and pepper' pattern of mixed clones in the center (*Appendix 1—figure 2B*). One of the parameters that can affect the sharpness of the patches' boundaries, in the coupled replication-removal case, is the size of the environment in which replication and removal are coupled, *m*. *Appendix 1—figure 2D* shows the dispersion parameter (*Corominas-Murtra et al., 2020*), as the function of time for different values of *m*. As *m* is lower, the borders are slightly sharper, yet the overall pattern remains the same.

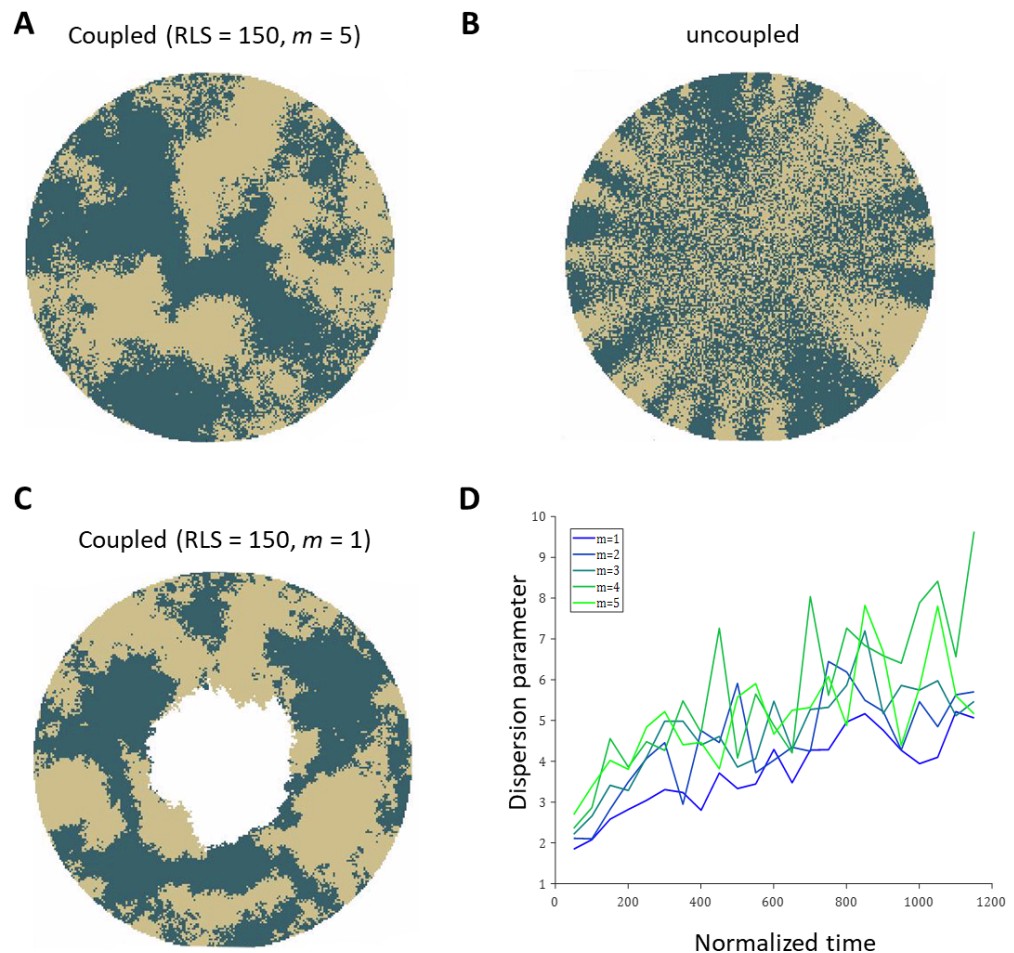

**Appendix 1—figure 2.** Realizations of the emerged patterns in the absence of centripetal bias.
(**A**) The case of coupled replication-removal with $m = 5$ and RLS = 150. The emerged pattern is that
of contiguous patches. (**B**) Uncoupled replication-removal with RLS = 4. (**C**) coupled replication-
removal with $m = 1$ and RLS = 150. The overall contiguous patches are similar to the $m = 5$ case.
Note that in this case, RLS = 150 is lower than the critical RLS needed to renew the
cornea. (**D**) Dispersion parameter as a function of normalized time for different values of m. Labeling
of the limbal stem cell occurs at time zero. As $m$ is smaller, the dispersion is slightly lower, yet the
overall pattern is similar.

## IV. Unmixing parameter

The are several parameters that capture clonal dispersion. For example, the parameter by *Coromi-
nas-Murtra et al., 2020* quantifies the dispersion of each clone by finding for each clonal cell the
maximal distance to its closest clonal neighbor. While useful, in this work we wanted to capture not
only clonal dispersion but also the deviation from a distribution where all the clones are perfect cir-
culars sectors with a base in the limbus and an apex in the center of the cornea ('centripetal stripes').
To quantify the clonal unmixing, we defined a parameter, $\phi$, that captures the deviation from this
'centripetal stripes' distribution (*Appendix 1—figure 3*). Consider a ring of cells (*Appendix 1—fig-
ure 3A*), in this ring, there could be several clones. The maximal angle between cells of a specific
same clone c is denoted by $\theta_c$, such that $0 \leq \theta_c \leq \pi$ (*Appendix 1—figure 3A*). This angle defines a
ring sector $\xi_c$. The unmixing parameter of the clone c in the $i^{th}$ ring is defined as,

$$\phi_{c,i} = \frac{\text{Number of cells of clone c in ring sector } \xi_c}{\text{Total number of cells in ringsector } \xi_c}. \tag{S1}$$

If all the cells in the sector $\xi_c$ are of from clone c, then, $\phi_{c,i} = 1$. If the sector $\xi_c$ contains different

clones, other than $c$, than $\phi_{c,i}$ is smaller than 1. $\phi_{c,i}$ is always positive and approaches zero and the number of cells from the clone $c$ is small, relative to other clones in the sector. The unmixing parameter of the $i^{th}$ ring is,

$$\phi_i = \left( \frac{1}{N_{c,i}} \sum_{c=1}^{N_{c,i}} \phi_{c,i} \right) \left( \frac{N_{c,i}}{N_{c,\text{limbus}}} \right), \tag{S2}$$

where $N_{c,i}$ is the number of clones in the $i^{th}$ ring, and $N_{c,limbus}$ is the number of clones in the limbus. The first term in *Equation S2* is simply the average of $\phi_{c,i}$ on the ring while the second term penalizes the case in which the number of clones in the ring decreases relative to the number of clones in the limbus. *Appendix 1—figure 2B* illustrates the value of $\phi$ for different patterns. In the case of a perfect 'centripetal stripes' pattern, $\phi_i$=1. In the case of mixed 'salt and paper,' pattern $\phi_i$ approaches zero. In the case of smooth patches that are not stripes, the unmixing parameter gets intermediate values. *Appendix 1—figure 3C* illustrates the dispersion parameter (*Corominas-Murtra et al., 2020*) for the different patterns that are shown in *Appendix 1—figure 3A*. The dispersion parameter is sensitive to the sharpness of the boundaries yet cannot differentiate between patches and stripes.

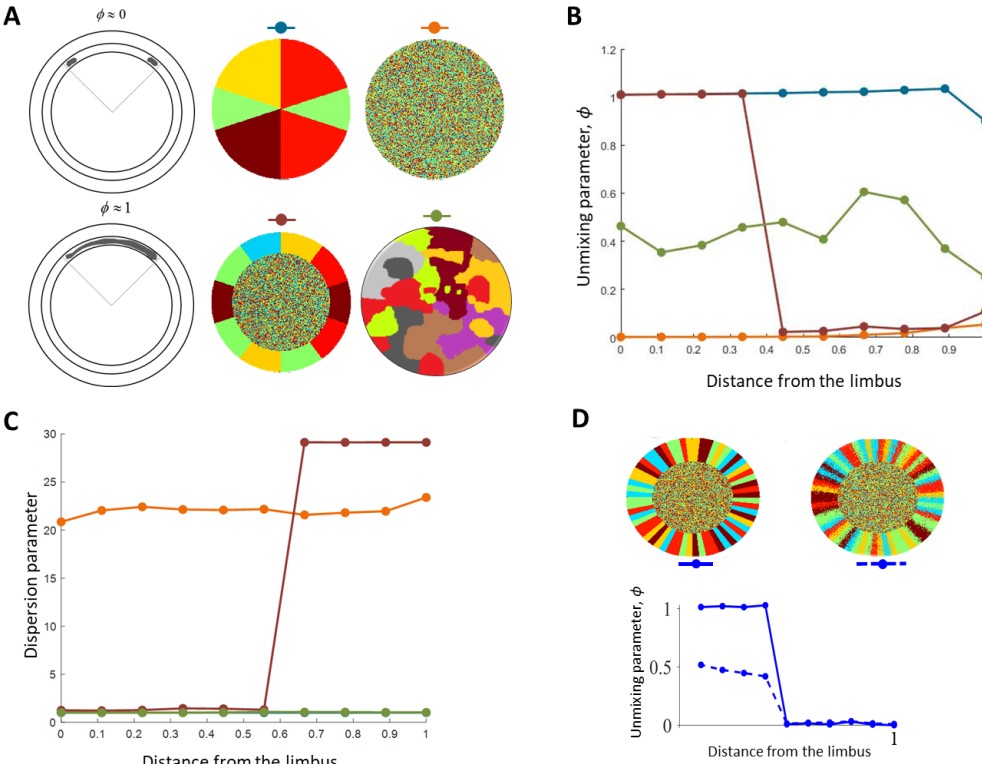

**Appendix 1—figure 3.** The unmixing parameter. (**A**) Example of different patterns. The color above each pattern is used in panels B and C. (**B**) The unmixing parameter as a function of the distance from the limbus for the different patterns in A. Patterns that have centripetal stripes (blue) get a high value compared to a mixed pattern (orange) or a patchy pattern (green). (**C**) The dispersion parameter for the different patterns in A. This metric is sensitive to the boundaries mixing and does not differentiate between patches and stripes (Note that the lines for the centripetal stripes (Blue) and patchy pattern overlay). (**D**) The unmixing parameter for ideal (left) and noisy(right) stripe pattern.

## V. Limits on coupled dynamics

### Renewal without bias

In this case, replication and removal occur at the same local environment of a circle with a radius of $m$ cells. As the cells are replicating, the lineage starts to propagate from the limbus towards the center. When $RLS = 1$, each corneal cell can replicate only once and therefore the maximal possible location of a removed cell is $m$ cells from the limbus. Thus, in steady-state, after many replications, the linage front will propagate to a distance of $m$ cells from the limbus.

When the RLS is increasing, the front location in steady-state will propagate further towards the center. Increasing the replicative lifespan by one will lead to a *maximal* increase of one cell in the front location (the deterministic limit). (*Figure 2B*). In the deterministic limit, the front location is equal to $m + RLS$ and thus the area of the renewed cornea scales as a $C - RLS^2$, where $C$ is a constant. The actual location of the front in the simulation is indeed closer to the limbus (*Figure 2B*). We captured the simulation trend by considering the propagation of the front as a stochastic process.

Consider a random variable, $x(t)$. At time interval, $\tau$,

$$x(t+\tau) = \begin{cases} x(t)+1 & p \\ x(t) & 1-p \end{cases} \tag{S3}$$

Taking $\tau = 1$, after $N$ generation $p(x(t+N) = \chi)$ is the same as binomial distribution; having $\chi$ successes out of $N$ trials, where $p$ is the probability of success.

If this process is being carried out many times, all possible locations, $\chi$, in which $p(x(t+N) = \chi) > 0$, will be populated. That is, we define the front location in steady-state as, $\chi_s = min(\chi | p(x(t+N) < \chi) = 1)$ which amounts to $\chi_s = min(\chi | 1 - p(x(t+N) < \chi) = 0)$.

In the case of the tissue, the number of trials is the replicative lifespan (RLS). The new cells that are formed can replace cells in a local neighborhood with a radius of $m$ cells. When there is no centripetal bias, the probability of success depends on the geometry of the grid and the radial shape of the front. Without bias, the probability of moving forward depends on the fraction of neighbors that are not reached by the clone yet. In the case of a rectangular grid, this fraction is 3/8. Using p=0.35 was used in the graph shown in *Figure 2B*.

### Limit on minimal RLS for renewal

We show in *Figure 3B* of the main text, that the minimal replicative lifespan required for renewal, $RLS_{min}$, decreases as a function of centripetal bias. Here we derive the theoretical lower limit on $RLS_{min}$ in the case of perfect bias.

As the front location from the limbus is smaller than $m$, the radius of the replication-removal area, the maximal propagation possible is doubling of the current distance from the limbus. Therefore, the minimal number of replication it will take the front to reach a distance $m$ from the limbus is log2 $(m)$. Once the front reached a distance $m$ from the limbus, the number of remaining replications is $RLS_{min} - \lceil log_2(m) \rceil$ and the maximal propagation of the front per one replication is $m$. Therefore the minimal number of replications to reach the center of the cornea is given by solving,

$$R = \lceil log_2(m) \rceil + (RLS_{min} - \lceil log_2(m) \rceil) \cdot m, \tag{S4}$$

and therefore,

$$RLS_{min} = \frac{R}{m} + \lceil log_2(m) \rceil \left(1 - \frac{1}{m}\right). \tag{S5}$$

In the case of *Figure 3B*, $m = 5$, $R = 100$ and therefore the limit on $RLS_{min}$ is around 22 replications.

### Limit on minimal renewal time

*Figure 3C* shows that as the centripetal bias is increased, the renewal time goes down. In the case of perfect bias, the average progress of the front is given by,

$$\langle s \rangle = \sum_{i=1}^{m} \frac{i}{m} = \frac{m+1}{2}. \tag{S6}$$

Therefore, the renewal, in this case, is simply, $R/\langle s \rangle = 2R/(m+1)$.

## VI. Limits on uncoupled dynamics

Renewal time dependence RLS. In *Figure 4C*, we show the dependence of the renewal time on RLS in the case of uncoupled replication-removal. First, we assume perfect bias and *RLS* = 0. In this case, only the stem cells replicate and the cells are removed from the center of the cornea. During $t_d$, the corneal doubling time, the number of stem cells which divide asymmetrically is given by, $(\lambda_s/\lambda_p) \cdot p_a \cdot 2\pi R$. If the front is located at a distance $R$ - $r$ from the limbus, the average number of advances per cell in the front per one corneal doubling time is

$$(\lambda_s/\lambda_p) \cdot p_a \cdot \frac{R}{r}. \tag{S7}$$

Thus, the time to propagate from the limbus to location $r$ is given by,

$$t_0 = \sum_{r=1}^{R} \frac{1}{(\lambda_s/\lambda_p) \cdot p_a \cdot \frac{R}{r}} = \frac{\lambda_p}{\lambda_s p_a} \left( \frac{R+1}{2} \right). \tag{S8}$$

In the case of $\lambda_p=1$, $\lambda_s=0.1$, $a=0.85$, $R = 100$, $t_0$ is around 600.

In the case there is no bias, the removed cell location is random. Thus, the probability that the front moves towards the center each time a stem cell divides depends on whether the removed cell is within the linage area or not. Therefore, the probability to propagate upon division is reduced by a factor of $r^2/R^2$ and *Equation S8* is changed to

$$t_0 = \sum_{r=1}^{R} \frac{1}{(\lambda_s/\lambda_p) \cdot p_a \cdot \frac{R}{r} \left( \frac{r^2}{R^2} \right)} = \frac{\lambda_p}{\lambda_s p_a} \cdot R \cdot H_R. \tag{S9}$$

where $H_R$ is the $R^{th}$ harmonic number. In the case that $\lambda_p=1$, $\lambda_s=0.1$, $a=0.85$ and $R = 100$, $t_0$ is around 6000.

As RLS is larger than zero, corneal cell replication also contributes to the advance of the front. At low RLSs, the number of cells that replicate increases by the factor of $2^{RLS}$, and thus the renewal time scales as $2^{-RLS}$ (*Figure 3C*).

### Stripe arrival time

In the case of perfect centripetal bias, in each division, cells are removed from the center of the cornea. Therefore, the maximal distance a stripe can advance in a single replication time is doubling its length and the minimal number of divisions needed for the stripe to reach the center is $log_2(R)$ (*Figure 5B*).

## VII. Velocity as a function of radial position

*Figure 7E* in the main text, which describes the front location as a function of normalized time, suggests that the velocity in which the front propagates depends on the radial location. *Appendix 1— figure 4* shows the propagation velocity ($dr/dt$) explicitly, for three different cases, as a function of radial position, $r$.

Close to the limbus ($r = 0$), in all cases, the front accelerates regardless of the model due to the radial boundary conditions. After $r \sim 10$, acceleration continues only if there is an inherent centripetal bias. In particular, the acceleration is significant in the case replication-removal is coupled. In this case of 'short-range' interaction, the front ring is getting smaller and thus the time to label them decreases. In the case of uncoupled replication-removal and RLS 4, most of the cornea is non-mitotic, and thus acceleration is impeded. In all cases with bias there is a deceleration near the center as the interaction of stripes through the center leads to clonal competition. In the case where there is no bias and no replication-removal coupling, there is a deceleration in the front velocity as the cells are farther from the boundary source.

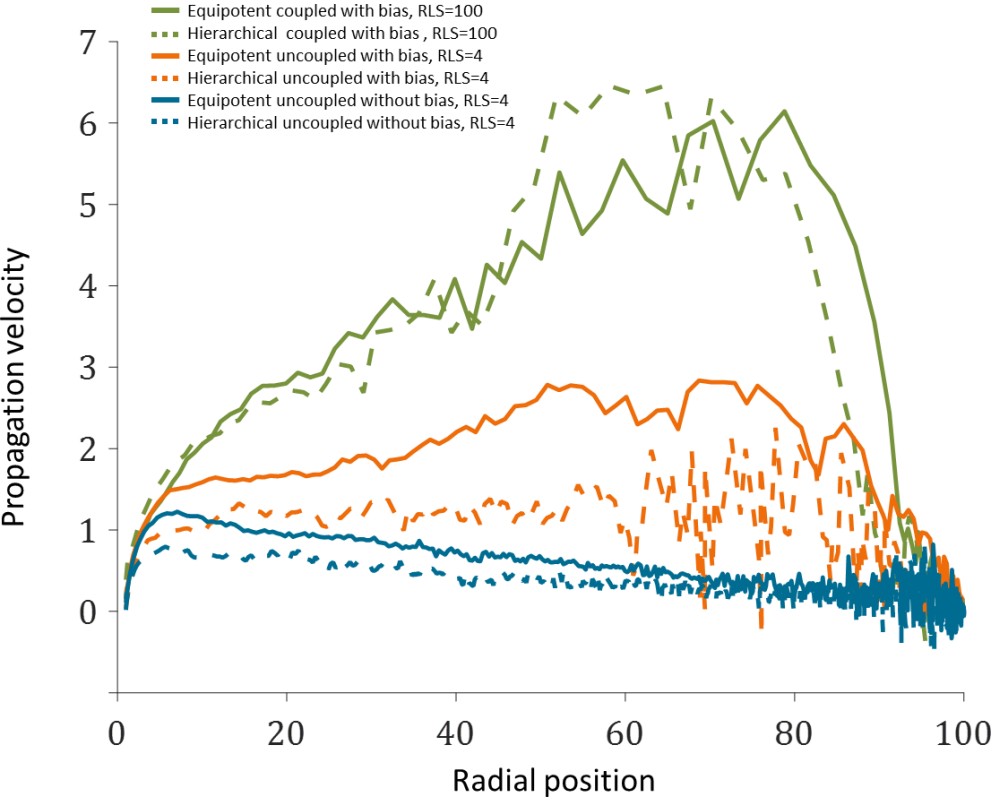

**Appendix 1—figure 4.** Propagation velocity as a function of radial position from the limbus. The data plotted is the mean of the three realizations shown in *Figure 7E* of the main text.

## VIII. Clones distributions

*Figure 7* in the main text shows how stem cell distribution affects the number of limbal and corneal clones as a function of time. The main predictions of the neutral drift clonal competition (which assumes that clones growth is based on diffusion) are that the number of clones declines as $1/\sqrt{t}$, the average clone size increases as $\sqrt{t}$, and that the distribution of clone size divided by the average ($n/<n>$) is time-invariant (in other words, clone size distributions scaled by the average will collapse onto the same curve) (*Klein and Simons, 2011*). The cornea geometry is different than the skin or the gut due to the unique circular geometry of the niche and tissue. Thus, clonal competition is affected by different, strict boundary conditions.

In the case of the Equipotent model, the dynamics of the limbal clone number and average limbal clone size are monotonically increasing and monotonically decreasing, respectively. Yet, they do not follow the neutral drift dynamics during the entire time trajectory. However, the limbal clone size distribution does exhibit scale invariance (*Appendix 1—figure 5*). In the case of the Hierarchical model, as expected, the clone number and average clone reach a plateau, and there is no scale invariance.

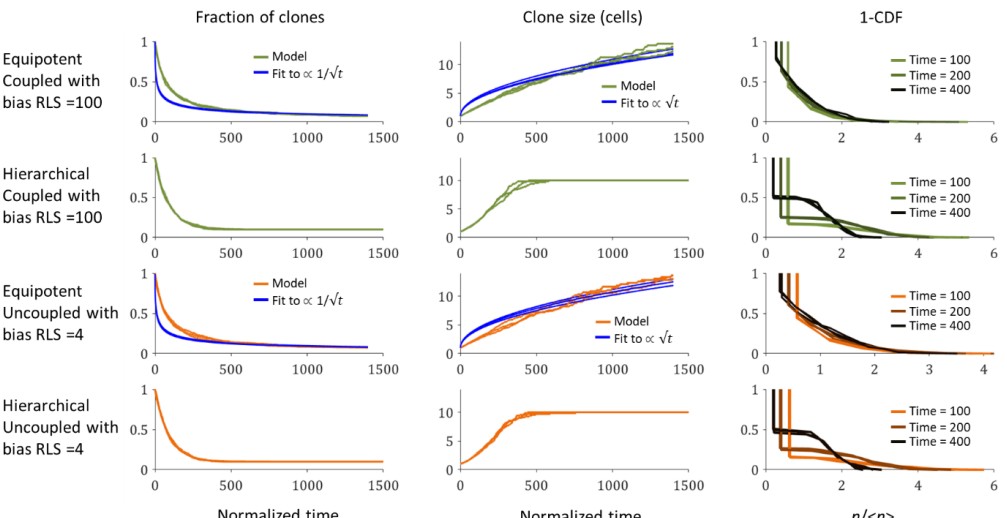

**Appendix 1—figure 5.** The dependence of limbal clone size distributions on time and stem cells properties. The fraction of clones as a function of time (left), the average clone size as a function of time (middle), and one minus the cumulative distribution function (CDF) of the clone size scaled by the average at three time points. Three realizations are shown. Blue lines are the best fit for the model.

## IX. Mutations and aging

The probability of mutation accumulation that results, for example, in cancer, depends on (i) the probability of acquiring oncogenic (or other deleterious) mutations, (ii) the number of proliferating cells, and (iii) the number of replication rounds in which mutations can happen before the mutated linage is removed. In the cornea, the number of cells is extremely low compared to the number of cells in the skin or the gut by a few orders of magnitude. Thus, just by considering the cell number, the probability of acquiring a particular set of mutations in the cornea is expected to be much smaller than the skin or the gut.

In the case of the Equipotent model, the mean time for a stem cell to be flushed out of the limbus is $t_s = \frac{1}{\lambda_s(1-p_a)}$. If there are no asymmetric replications, $p_a = 0$, the typical flush out rate is the typical stem cell replication time. If there are no symmetric replications, $p_a = 1$, stem cells in the limbus are not replenished. Thus, mutations tend to be flushed out faster as there are more symmetric replications. In the cornea, all the replications are symmetric, and thus the mean dwell time is $t_c = \frac{1}{p_a\lambda_s(N_s/N_p)+\lambda_p} \approx \frac{1}{\lambda_p}$. In the case of the Hierarchical model, once a mutation occurs in the stem cells it cannot be flushed out, and the clone with this mutation can continue to accumulate mutations.

In the Equipotent case, once a mutation occurs, it has a probability of $\frac{1-p_a}{2-p_a}$ to be removed within one replication. Thus, in our case, about 15% of the mutations will not leave the cornea. The limbal cells that were able to lunch a clone to the cornea have a lifetime distribution with a median of around 100 replication times in the case of the coupled model with bias, around 130 replication times in the case of uncoupled with bias, and around 270 replication times in the case of uncoupled without bias (*Appendix 1—figure 6*). It is important to note that linage lifetime does not indicate the maximal number of divisions that the linage underwent. The limit on the maximal depth of a linage is shown in *Appendix 1—figure 6*, and its median is in the order of 100 replications in the case of the coupled model with bias, around 20 replications in the case of uncoupled with bias, and around 30 replications in the case of uncoupled without bias.

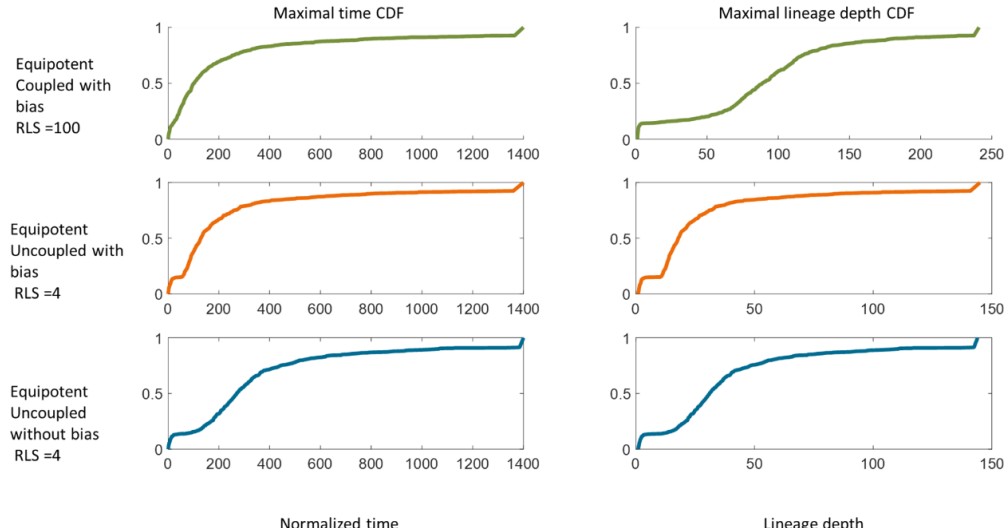

**Appendix 1—figure 6.** The cumulative probability distribution of linage removal time (left) and linage depth (right) for different models.

# X. Model parameters

**Appendix 1—table 1.** Model parameters and references.

| Parameter | Value and references |
|---|---|
| The radius of the cornea and basal corneal epithelial cell, $R$. | 100 Cells. (**Di Girolamo et al., 2015**; **Dorà et al., 2015**) |
| Probability of asymmetric replication of $S$ cells, $p_a$. | 0.85, Cornea (**Richardson et al., 2016**), Epidermis (**Mascré et al., 2012**), Esophagus (**Doupé et al., 2012**) |
| Neighborhood radius of interacting cell in the 'coupled' model, $m$. | 5 cells (**Mesa et al., 2018**; **Miroshnikova et al., 2018**) |
| Normalized renewal time. | Estimated cell cycle times based on double DNA labeling: Limbal label retaining cells ~ 14–21 days Limbal area cells (stem and early progenitors), Peripheral cornea, Central cornea ~ 3.0–5.5 days (**Sagga et al., 2018**). All of our time estimations are normalized to the doubling time of the cells. We used a relative proliferative rate $(\lambda_s/\lambda_p)$ that equals to 0.1. |
| Relative proliferative rates of the stem cells and progenitor cells $(\lambda_s/\lambda_p)$. | 0.1, (**Lavker et al., 1991**; **Sartaj et al., 2017**) |
| The fraction of stem cells inside the limbus in the Hierarchical model, $f_s$. | In the Hierarchical model estimates using label retaining vary from 3% to 20% (**Lavker et al., 1991**; **Sagga et al., 2018**; **Sartaj et al., 2017**; ) we used 10%. |

