## [Decision Letter]

**Acceptance summary:**

Understanding homeostasis of tissues renewed by localised populations of stem cells requires the development of models to simulate the dynamics of cellular turnover. Savir etal. propose a 2D lattice-based Monte-Carlo model of stem cell renewal of the cornea, which is an interesting model system as it is renewed by localised populations of stem cells, requiring long-range flows to balance cell loss. They incorporate the main relevant features that include the modes of stem cell divisions (stochastic vs. deterministic), the spatial correlations between the locations of replication and cell removal, the directionality bias of the division planes and the replicative lifespan of progenitor cells. This model provides a general relationship between spatial correlations between duplication and removal and the replicative lifespan (number of cell division before death) of stem cells. The modelling approach is sufficiently generic so this this model should apply to other tissues with similar competitive dynamics. In the case of the cornea, the model predicts that the replicative lifespan is sufficient to allow the complete rejuvenation of the tissue, and that the spatial distribution of stem cells in the periphery of the cornea does not affect the competition between cell duplication and removal.

**Decision letter after peer review:**

Thank you for submitting your article "The role of replication‐removal spatial correlations and cellular replicative lifespan in corneal epithelium homeostasis" for consideration by *eLife*. Your article has been reviewed by Aleksandra Walczak as the Senior Editor, a Reviewing Editor, and two reviewers. The following individuals involved in review of your submission have agreed to reveal their identity: Shalev Itzkovitz (Reviewer #1).

The reviewers have discussed the reviews with one another and the Reviewing Editor has drafted this decision to help you prepare a revised submission.

Summary:

Understanding homeostasis of stem cell maintained tissues requires the development of models to simulate the dynamics of cellular turnover. In this manuscript, the authors propose a 2D computational model of stem cell renewal of the cornea, which is an interesting model system as it is renewed by localised populations of stem cells, requiring long-range flows to balance cell loss. They incorporate the main relevant features that include the modes of stem cell divisions (stochastic vs. deterministic), the spatial correlations between the locations of replication and cell removal, the directionality bias of the division planes and the replicative lifespan of progenitor cells.

The reviewers and myself find your work interesting and your approach elegant. Both the methodology and the results will be of interest to both systems biologists and cell biologists interested in the modelling of tissue dynamics. The geometry of the cornea system simulated is very interesting and leads to non-trivial results that really require this kind of computational approach. The paper is clearly written and the methods and theoretical approach well described.

However, the paper would strongly benefit from more links to experiments and to the existing literature. We also ask you to consider alternative hypotheses, in particular those involving spatial coupling through mechanical effects.

Please consider the list of comments below to help you revise your manuscript and increase the generality and impact of your work.

Essential revisions:

1) The title right now is quite general, could the title be reformulated to emphasise the findings (for instance on the tradeoffs, or the impact on clonal shapes).

2) The non expert readers would benefit from a better description of the experimental system you are modelling. This appears to be a lineage tracing mouse model for exploring clonal dynamics. Are all cells in the cornea labelled or rather only the stem cells? Is this a confetti mouse system? Could it be possible to reproduce some of the published experimental data next to your numerical results in a Supplementary information figure. This would be quite visual and demonstrative for readers not directly familiar with the system.

3) There addition of experimental data would strengthen your work.

(3.1) Are there measurements of EdU or BrdU incorporation in the cornea that could validate some of the model predictions, e.g. the radial positions where cells proliferate as well as the rates of proliferation? What are these rates (once per week? Once per month)? If such data exists it should be added to Table S1.

– Are there estimates of the number of stem cells per cornea? If yes, please add to Table S1.

(3.2) Can the velocity of cells as a function of radial position be analyse? If all cells divide with a complete radial bias, that cell movement would be accelerated as cell approach the centers, similarly to the situation in the intestinal crypts, where the higher cells are along the crypt axis the faster they move (e.g. PMID 28049136).

(3.3) Could experimentally measured value of the fraction of post-mitotic cells be added to Figure 6B.

4) Further comparison with existing literature is required.

(4.1) Regarding the spatial structure of the patches. You mention

Findlay et al., 2016; Kucerova et al., 2012; Mort et al., 2011; Douvaras et al., 2013;, where messy patches instead of continuous strips are observed, and use it to motivate your simulations in the condition of lost directionality. This is a nice idea, but looking at the references, it looks like in these mutants, patches are still incredibly sharp and distinct from one another, with really little dispersion (for instance Pax6 mutants). This is visually quite different from what is shown for instance on Figure 2. In fact, even for high directionality (Figure 3), stripes look much more messy than the experimental counterparts.

Does this mean that the spatial coupling should be smaller than the current value (m=5), which is based on a different system (epidermis in Mesa et al., 2018; Miroshnikova et al., 2018). Exploring this further would be valuable.

– The unmixing parameters seems quite influenced by the 1D stripe geometry: for instance, the green curve in Figure S1A shows fairly low value of \phi although it corresponds to very sharp and unmixed domains (from a 2D perspective). It is therefore important and helpful to experimentalists to quantify the clonal boundary roughness differently. Some work have explored the effect of stochastic clonal dispersion/competition on clone shape and size (for instance Rulands et al., 2018 as well as Corominas-Murtra et al., and Hallatschek et al., 2007 in the presence of net flows/expansion), which could have some useful theoretical formula/connections to this.

(4.2) You should discuss more in detail Moraki, Grima and Painter, 2019, which models the same system of cornea. You only mention it to say that it was a 1D model – and the 2D aspect is an important step forwards – but Moraki, Grima and Painter, 2019 did look at the influence of replication cycle, fraction of stem cells and division rates for cornea renewal, so it's necessary to compare and contrast in the Discussion.

(4.3) You explore the evolution of clone number as a function of time in Figure 7 but do not comment on the shape of these distributions, which have been derived analytically for a number of systems and geometries (Klein and Simons, 2011 for instance). Competition across a 1D ring in the limbus for instance is expected to give rise to gaussian distributions of clone sizes, average clone sizes increasing as sqrt(t) and number of clones decreasing as 1/sqrt(t). You should check if your distributions follow this expectation?

5) Alternative models could be considered, for instance removal driving the system. You only consider unidirectional correlations (division -> removal). But the reverse could be possible. Directional stripes could result from cells dying in the center of the patch, creating a negative pressure that would drive a centripetal flow of cell (in the hydrodynamic sense). In general, mechanics is entirely absent from the algorithm at present. Could it be included in the model, or at least discussed in a more quantitative way.

6) The Discussion already mentions the relation to pathologies and in particular cancer. It could be interesting to expand this a bit in relation with the proposed model. For instance, is there cancer of the cornea? If not, might this indicate optimality of the system to avoid accumulation of oncogenic mutations? How do the different models affect the accumulations/flushing of mutations in lineages with time? Purely asymmetric divisions can lead to higher accumulation of a series of oncogenic mutations whereas symmetric stem cell divisions can stochastically flush out oncogenic mutations. One may assume that the extent of replication and the geometry could also affect the “depth” of each lineage and consequently the numbers of mutations it would accumulate before oncogenic transformation. You could refer to PMID 14988930, 24264992 when discussing this point.

7) It would be nice to add the simulation code as a supplement.

---

## [Author Response]

Essential revisions:1) The title right now is quite general, could the title be reformulated to emphasise the findings (for instance on the tradeoffs, or the impact on clonal shapes).

Following the reviewer's remark, we changed the title to “Spatial correlations constrain cellular lifespan and pattern formation in corneal epithelium homeostasis”.

2) The non expert readers would benefit from a better description of the experimental system you are modelling. This appears to be a lineage tracing mouse model for exploring clonal dynamics. Are all cells in the cornea labelled or rather only the stem cells? Is this a confetti mouse system? Could it be possible to reproduce some of the published experimental data next to your numerical results in a S.I. figure. This would be quite visual and demonstrative for readers not directly familiar with the system.

We thank the reviewer for this important comment. Current linage tracing experimental setups are indeed based on the confetti mouse system. In these mice (such as double transgenic K14-Cre^ERT2^;Brainbow^2.1^), injection of tamoxifen results in a stochastic, irreversible, expression of fluorescent markers. The expression can occur in all the K14+ basal cells in the limbus or corneal center. With time, radial stripes that emerge from the limbus can be distinguished, and their kinetics can be measured. Moreover, with time, these stripes constitute the majority of the labeled area (Amitai-Lange et al., 2015).

Following the reviewer's comment, we added a section to the Supplementary information (Appendix 1 Section I) that elaborates on the experimental system. We added to this section a figure with fluorescence microscopy photomicrograph of a cornea from a double transgenic K14-Cre^ERT2^;Brainbow^2.1^ mice side by side with a snapshot of the model (Appendix 1 Figure S1).

3) There addition of experimental data would strengthen your work.(3.1) Are there measurements of EdU or BrdU incorporation in the cornea that could validate some of the model predictions, e.g. the radial positions where cells proliferate as well as the rates of proliferation? What are these rates (once per week? Once per month)? If such data exists it should be added to Table S1.

We thank the reviewer for this valuable comment. Recent nucleotide analogue (BrDU, EdU, IdU) incorporation “pulse-chase” experiments demonstrate that corneal cells proliferate in all radial positions with approximately the same rate of once per 3-5 days in mice (Sagga et al., 2018). Together with the estimation for total corneal renewal time, which is in the order of a few months (Amitai-Lange et al., 2015), it gives an estimate for the propagation time to fully renew the cornea of 20 to 50 corneal cell doubling times. Thus, in our manuscript, we used a normalized renewal time of 100 cell doubling as an upper limit.

As the reviewer suggested, we added these values to Table S1 and elaborate on these experimental limits in the manuscript in the Results' section.

– Are there estimates of the number of stem cells per cornea? If yes, please add to Table S1.

The number of stem cell estimations differs according to the stem cell distribution model (Equipotent or Hierarchical). In the hierarchical model, the fraction of limbal stem cells out of the total number of limbal cells (which is in the order of 600 cells) varies from 3% to 20% (Sagga et al., 2018),(Sartaj et al., 2017). Following the reviewer's comment, we added these estimates to Table S1.

(3.2) Can the velocity of cells as a function of radial position be analyse? If all cells divide with a complete radial bias, that cell movement would be accelerated as cell approach the centers, similarly to the situation in the intestinal crypts, where the higher cells are along the crypt axis the faster they move (e.g. PMID 28049136).

For thank the reviewer for this intriguing comment. Figure 7E (in both the revised and original manuscript), which describes the front location as a function of normalized time, indeed suggests that the velocity in which the front propagates depends on the radial location. A concave curve indicates that the front velocity accelerates as it is closer to the center, while a convex curve suggests a slowdown. To explicitly show it, we added a section to the Supplementary information (Section VII. Velocity as a function of radial position) and a figure, Figure S4, which shows the propagation velocity, for the different models, as a function of radial position. The results suggest that in the case of centripetal bias (which is similar to the intestine case), the velocity of the front propagation is indeed accelerating (till it gets to a distance of few cells away from the center and slows down due to the circular pushes from all directions). We elaborate more on these findings and cite the relevant reference on this interesting phenomenon in the main text in the Results section.

(3.3) Could experimentally measured value of the fraction of post-mitotic cells be added to Figure 6B.

We thank the reviewer for this important comment. It is not trivial to infer the exact fraction of post-mitotic cells in the cornea. Yet, accumulating evidence suggests that the case in which the majority of the center of the cornea is post-mitotic is highly unlikely. Double labeling (IdU-EdU) experiments that estimate S-phase/cell cycle ratio, together with IdU “short pulse” experiments that measure the labeling index, suggest that most of the cells in all area of the cornea are actively mitotic (Sagga et al., 2018) (Richardson et al., 2017). This is consistent with “long chase” experiments (Li et al., 2017) that resulted in a small fraction of label-retaining cells in the central cornea.

These results are indeed important in the context of Figure 6B where we discuss the fraction of the post-mitotic cell of each different case. Following the reviewer's comment, we added subsection “Limits on different models” in the main text that discusses these experimental observations and their relation to the model.

4) Further comparison with existing literature is required.(4.1) Regarding the spatial structure of the patches. You mentionFindlay et al., 2016; Kucerova et al., 2012; Mort et al., 2011; Douvaras et al., 2013;, where messy patches instead of continuous strips are observed, and use it to motivate your simulations in the condition of lost directionality. This is a nice idea, but looking at the references, it looks like in these mutants, patches are still incredibly sharp and distinct from one another, with really little dispersion (for instance Pax6 mutants). This is visually quite different from what is shown for instance on Figure 2. In fact, even for high directionality (Figure 3), stripes look much more messy than the experimental counterparts.Does this mean that the spatial coupling should be smaller than the current value (m=5), which is based on a different system (epidermis in Mesa et al., 2018; Miroshnikova et al., 2018). Exploring this further would be valuable.

Following the reviewer's comment and to aid the reader in comparing the emerged patterns in the cases there is no centripetal bias, we added a section to the Supplementary information (Appendix 1 Section III) with realizations in which there is only one color that label the cells (Appendix 1 Figure S2). This representation, which is closer to the experimental setting of the mutant experiments, highlights the inherent qualitative difference between the emerged patterns in the coupled and uncoupled cases. In particular, no contiguous patches can emerge in the uncoupled case.

We thank the reviewer for the intriguing comment regarding the effect of the interaction range. Following the reviewer's comment, we analyzed the boundary dispersion for different values of *m* (Appendix 1 Figure S2 in the revised Supplementary information). Reducing the value of *m* resulted in slightly less dispersed boundaries, yet the overall pattern remains similar.

– The unmixing parameters seems quite influenced by the 1D stripe geometry: for instance, the green curve in Figure S1A shows fairly low value of φ although it corresponds to very sharp and unmixed domains (from a 2D perspective). It is therefore important and helpful to experimentalists to quantify the clonal boundary roughness differently. Some work have explored the effect of stochastic clonal dispersion/competition on clone shape and size (for instance Rulands et al., 2018 as well as Corominas-Murtra et al., and Hallatschek et al., 2007 in the presence of net flows/expansion), which could have some useful theoretical formula/connections to this.

We thank the reviewer for this valuable comment. As the reviewer mentioned, our unmixing parameter indeed depends on the patches' geometry. This is by design as the main geometrical order property we wish to capture is the centripetal stripeness of the pattern (how much it resembles a “pizza” like shape) rather than the disorder of the boundaries. Our metric can capture both the stripe geometry and the noise resulted from inter-clonal mixing. Both the metrics mentioned by the reviewer (and in particular, Corominas-Murtra et al.,) cannot distinguish between a patch-like pattern and a stripe-like pattern with the same boundary sharpness.

Following the reviewer's comment, we added a figure to the Supplementary information (Appendix 1 Figure S3C) where we compare the unmixing parameter with the boundary dispersion metric and emphasize this point in the revised Supplementary information ( Section IV. Unmixing parameter).

(4.2) You should discuss more in detail Moraki, Grima and Painter, 2019, which models the same system of cornea. You only mention it to say that it was a 1D model – and the 2D aspect is an important step forwards – but Moraki, Grima and Painter, 2019 did look at the influence of replication cycle, fraction of stem cells and division rates for cornea renewal, so it's necessary to compare and contrast in the Discussion.

We thank the reviewers for this comment. Marki et al., analyzed the 1D master equation that considers the symmetric/asymmetric replication of stem cells into transient amplifying cells that, after a limited number of replications, produce terminally differentiated cells. This model does not account for spatial interactions and bias, or different stem cells spatial distributions. It is a 1D approximation of our Equipotent uncoupled model without bias (Equally distributed stem cells and no spatial correlation between replication and removal, no centripetal bias). The authors analyze the effect of replication time and replication lifespan on the mean cell number and their second order fluctuations. The relevant implication of their analysis to the physiological case is that the RLS that provides homeostasis is 4-12 replications, and that for this range of RLS, fluctuations can be maintained under some conditions. It is insightful to compare these results to the effect of RLS on a private case of our Equipotent uncoupled model which also suggests that in this range of RLS, renewal times are minimal.

Following the reviewer remarks, we discuss how this model is a private case of our results in the Results section and in the Discussion section.

(4.3) You explore the evolution of clone number as a function of time in Figure 7 but do not comment on the shape of these distributions, which have been derived analytically for a number of systems and geometries (Klein and Simons, 2011 for instance). Competition across a 1D ring in the limbus for instance is expected to give rise to gaussian distributions of clone sizes, average clone sizes increasing as sqrt(t) and number of clones decreasing as 1/sqrt(t). You should check if your distributions follow this expectation?

We thank the reviewer for this intriguing comment. It is indeed insightful to compare our results, and in particular, the Equipotent model to neutral drift clonal competition in which the main predictions are that the number of clones declines as , the average clone size increases as , and the distribution of clone size divided by the average (N) is time-invariant (in other words, clone size distributions scaled by the average will collapse onto the same curve). In the Equipotent case, the dynamics of the limbal clone number and average limbal clone size are monotonically increasing and monotonically decreasing, respectively. Yet, they do not follow the neutral drift dynamics during the entire time trajectory. However, the limbal clone size distribution exhibits scale invariance (Supplementary information Appendix 1 section VII. Velocity as a function of radial position, Appendix 1 Figure S5). In the Hierarchical case, as expected, the clone number and average clone size reach a plateau, and there is no scale invariance.

Following the reviewer's comment, we discuss this result in the main text (subsection “The effect of stem cells distribution and dynamics”) and added a section to the Supplementary information (Appendix 1 Section VII. Velocity as a function of radial position, Figure S5).

5) Alternative models could be considered, for instance removal driving the system. You only consider unidirectional correlations (division -> removal). But the reverse could be possible. Directional stripes could result from cells dying in the center of the patch, creating a negative pressure that would drive a centripetal flow of cell (in the hydrodynamic sense). In general, mechanics is entirely absent from the algorithm at present. Could it be included in the model, or at least discussed in a more quantitative way.

We thank the reviewer for this important comment. As the reviewer noted, similar patterns can emerge regardless of the temporal order of replication and removal. Our model does not assume a temporal order between replication and removal and is not unidirectional. In each step of the simulation, the pair of cells involved in the replication-removal process is chosen together according to the assumed correlations. That is, there is no preference for whether replication or removal happens first. Once the locations of the “hole” and the duplicated cell are determined, mechanics play a major role in the algorithm as they dictate the reorientation of the cells after duplication and removal.

The spatial correlations in the model represent the effective result of mechanical (or chemical) interactions between the cells. In this work, we use two types of interactions that represent two limits: short-range interactions with a typical interaction length of *m* cells (“coupled”), and long-range interactions where all cells interact with each other (“uncoupled”).

Following the reviewer's remark, we highlight the relation between the mechanical interactions and the effective correlation in the Discussion section of the revised manuscript. In particular, we discuss the quantitative effect of changing the interaction range *m* on the results (Discussion). We also revised the methods section in the main text (Materials and methods) and section II in the Supplementary information to highlight this important feature of the model.

6) The discussion already mentions the relation to pathologies and in particular cancer. It could be interesting to expand this a bit in relation with the proposed model. For instance, is there cancer of the cornea? If not, might this indicate optimality of the system to avoid accumulation of oncogenic mutations? How do the different models affect the accumulations/flushing of mutations in lineages with time? Purely asymmetric divisions can lead to higher accumulation of a series of oncogenic mutations whereas symmetric stem cell divisions can stochastically flush out oncogenic mutations. One may assume that the extent of replication and the geometry could also affect the “depth” of each lineage and consequently the numbers of mutations it would accumulate before oncogenic transformation. You could refer to PMID 14988930, 24264992 when discussing this point.

We thank the reviewer for this interesting comment. Cancer in the cornea is relatively rare, and when it occurs, it is believed to originate from the conjunctiva or the limbus. The probability of mutation accumulation that results, for example, in cancer depends on (i) the probability of acquiring oncogenic (or other deleterious) mutations, (ii) the number of cells, and (iii) the number of replication in which mutations can happen before the mutated linage is removed. In the cornea, the number of cells is extremely low compared to the number of cells in the skin or the gut, for example. Thus, just by considering the cell number, the probability of acquiring a particular set of mutations is expected to be significantly smaller than the skin or gut. Another fascinating aspect that the reviewer mentioned, is that symmetric divisions promote cell exchange and can efficiently flush out mutations. Therefore, the Equipotent model is more resilient to mutations in the limbus.

Following the reviewer's comment, we added a section to the Supplementary information (Section IX. Mutations and aging) where we calculate the time of a linage originated from a mutated cell to be completely removed from the cornea, and the limit on the maximal depth of these linages. Overall, centripetal bias decreases the depth of the linage and the radius of the cornea imposes a limit on the median depth. We also discuss these results in the Discussion section in the revised manuscript.

7) It would be nice to add the simulation code as a supplement.

We thank the reviewer for this important comment. We uploaded our simulation code (written in MATLAB) to an open software archive as requested https://github.com/SavirLab/CorneaSim